# WHAT TIME TELLS US? TIME-AWARE REPRESENTATION LEARNING FROM STATIC IMAGES

## ABSTRACT

Time becomes visible through changes in what we see, as daylight fades and shadows grow. Inspired by this, in this paper we explore the potential to learn time-aware representations from static images, trying to answer: *what time tells us?* To this end, we first introduce a Time-Oriented Collection (TOC) dataset, which contains 130,906 images with reliable timestamps. Leveraging this dataset, we propose a Time-Image Contrastive Learning (TICL) approach to jointly model timestamp and related visual representations through cross-modal contrastive learning. We found that the proposed TICL, 1) not only achieve state-of-the-art performance on the timestamp estimation task, over various benchmark metrics, 2) but also, interestingly, though only seeing static images, the representations learned by TICL show strong capability in several time-aware downstream tasks such as time-based image retrieval, video scene classification, and time-aware image editing. Our findings confirm that time-aware visual representations are learnable from static images and beneficial for various vision tasks, laying a foundation for future research on understanding time-related visual context. [1]

> "Time is the moving image of eternity."
>
> *Plato*

## 1 INTRODUCTION

On our planet, the day-night cycle occurs every 24 hours, a phenomenon recorded systematically by various clock systems developed by human society. Surprisingly, such clock systems emerged much earlier than our recognition of Earth as a "blue marble" engaged in constant orbital movement (Dohrn-van Rossum, 1996). Although most people possess a vague, intuitive sense of current time (Moore, 1992), the origin of this metaphysical consciousness of time, which is a key concept for both our bodies and society, remains elusive. Research in neuroscience has revealed that visual stimuli from photoreceptors are crucial for the adaptation of mammals to day-night rhythms (Duffy & Czeisler, 2009). This implies that the concept of time for humankind could emerge from various visual experiences. Given the implicit relations between clock time and visual experiences, we are interested in asking:

- *Can neural networks learn a representation of time from visual stimuli i.e. static images?*

- *If so, what implications does such representation tell us towards understanding the world?*

To answer these questions, in this study, we propose an approach to learn and disentangle the time representations from static images, by estimating timestamps from images. There are previous attempts trying to address this task using simple datasets with fixed views, such as the Time of Year Dataset (TYD) (Volokitin et al., 2016) and other subsets of the Archive of Many Outdoor Scenes (AMOS) (Jacobs et al., 2009), featuring images captured by a few stationary webcams at different times of the day. However, we found these datasets do not reflect the complexity and diversity of views in real-world applications. Salem et al. (2020) proposed a mixed subset of AMOS and

---

[1]Code and dataset are available at: `https://anonymous.4open.science/r/TICL-6E2F`

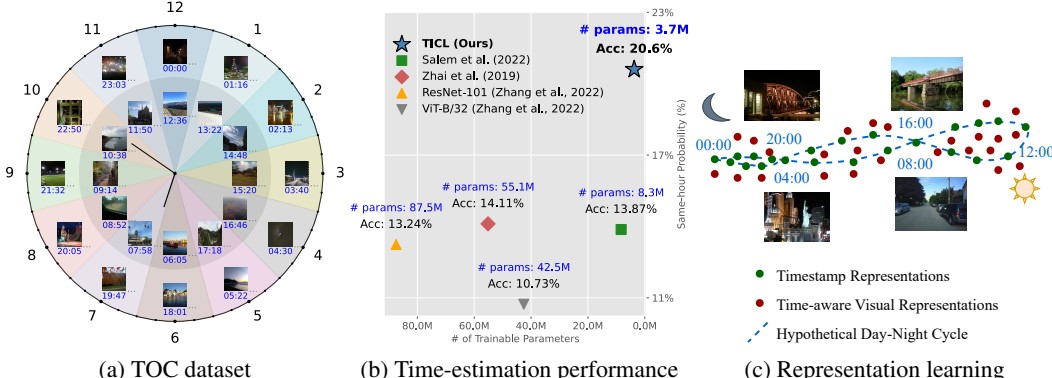

(a) TOC dataset  (b) Time-estimation performance  (c) Representation learning

Figure 1: **An overview of our study,** in which we presented a new high-quality dataset for timestamp estimation (a), based on which we propose a new approach, achieving state-of-the-art performance (b). We further analyse the representations learned through this process (c), showing effectiveness over several time-related downstream tasks.

YFCC100M (Thomee et al., 2016), but it contains many images with incorrect timestamps due to unsynchronised time zones Padilha et al. (2022), undermining its reliability to learn time-aware representations from.

In addition to the challenge of lacking reliable datasets, the algorithms proposed for this task also face significant difficulty. As reported in Volokitin et al. (2016); Sharma et al. (2016); Zhang et al. (2022), there are inherent ambiguities between the ground truth timestamp and images due to the dependence of daylight time on other metadata (*e.g.* regions and seasons). To cope with this issue, Salem et al. (2022); Zhai et al. (2019) introduce additional input such as other metadata, aiming to model the conditional probabilities between geolocation, hour, month, and even week of the year to provide performance improvements to the task itself. While these efforts made reasonable and valuable explorations, they have introduced extra dependencies on additional metadata, limiting the generalisation ability when such metadata is unavailable. On the other hand, they only focus on the specific task of timestamp estimation, without looking into the learned representations. Whereas in this work, in addition to estimating more accurate timestamps, we further investigate the learned time-aware representation and show their effectiveness in several downstream tasks.

Specifically, due to the lack of high-quality data, we first collect a new benchmark dataset comprising social media images featuring diverse views and objects, along with manually verified reliable timestamps. Such a dataset has the potential to become the new de facto choice for future research. Secondly, regarding the issues that emerged in previous works, we propose a Time-Image Contrastive Learning (TICL) approach to jointly model visual and time representations for accurate timestamp estimation. The proposed TICL approach outperforms all existing methods for timestamp estimation. Moreover, we also conduct validations of the learned time-aware representations on several downstream tasks, including time-based image retrieval, video scene recognition, and time-aware visual editing, showing the effectiveness of the learned representations. Our key contributions can be summarised as follows:

- We introduce a Time-Oriented Collection (TOC) dataset, a new benchmark dataset that contains 130,906 images with reliable timestamps (examples shown in figure 1a).

- We propose TICL, an approach to jointly model time and related visual representations, achieving state-of-the-art (SOTA) performance on timestamps estimation from static images. Figure 1b shows the performance achieved, boosting SOTA from **14.11%** to **20.6%**, while keeping small number of trainable parameters.

- We study the learned time-aware visual representations (figure 1c) by validating them on several downstream tasks, showing clear evidence of their effectiveness, in tasks such as time-based image retrieval, video scene recognition, and time-aware image editing.

## 2 RELATED WORKS

### 2.1 TIME OF DAY ESTIMATION

Estimating the time of day from static images presents a notable challenge, a topic that remains largely underexplored in the literature. Earlier studies were hampered by the scarcity of datasets with images paired with accurate local timestamps. Many images, particularly those sourced from social networks, contain metadata that is often inaccurate, lost, or not calibrated to the local time-zone. To cope with this, some researchers have turned to webcam image datasets, which naturally include accurate timestamps. However, these datasets are typically limited to fixed singular views and are frequently degraded by environmental noise and adverse conditions like low light or physical obstructions. Such limitations hinder the generalisation of the time estimation model from webcam data to more diverse applications.

For example, established social media image datasets such as MIRFLICKR-1M (Huiskes & Lew, 2008) and YFCC100M (Thomee et al., 2016) include only a small fraction of images with reliable metadata. On the other hand, webcam datasets contain only fixed stationary views, such as AMOS (Jacobs et al., 2007) and TYD dataset (Volokitin et al., 2016), which fail to represent the complexities of temporal variations within diverse environments. The CVT-Time dataset (Salem et al., 2020), despite combining stationary webcam images with more diverse YFCC100M subsets, still struggles with unreliable timestamps and low-quality webcam images.

Existing time estimation methods have primarily focused on classifying images into time periods. Sharma et al. (2016) addressed this with a small social media image dataset using SVMs and early CNNs, classifying images into four broad time classes, which oversimplified the task. Volokitin et al. (2016) used VGG-16 to predict temperature, month, and hour from images taken by 6 webcams during daylight, which is insufficient for comprehensive day-long analysis. Zhang et al. (2022) evaluated baseline models, such as He et al. (2015) and Dosovitskiy et al. (2021), on images from a single webcam. Zhai et al. (2019) worked with a mixed dataset of Flickr and webcam images, classifying images taken at the same hour but in different months into 288 classes, optionally incorporating geolocation inputs. Similarly, Salem et al. (2022) used webcam images, predicting month, week, and hour as dependent tasks trained jointly while considering geolocations as optional inputs.

### 2.2 TIME-AWARE REPRESENTATION LEARNING

Although there has been limited focus specifically on learning time-aware visual representations, related studies have explored visual attributes based on human annotations. An early attempt in this direction is the transient attribute database (Laffont et al., 2014), which aimed to derive attribute-based representations from static images. This database contains images from 101 stationary cameras, with each image annotated by human evaluators for 40 numerical transient attributes. Notably, some of these attributes, such as *sunrise*, *night*, and *daylight*, are inherently tied to the time of day. However, the dataset lacks precise timestamps for the images, and the attributes rely on subjective human judgements, which limits the scalability and generalisation of the approach.

A subsequent work aimed to bridge the gap between image metadata and scene attributes (Salem et al., 2020). This study explored the dynamic mapping of visual appearances over time and across geolocations. Specifically, given a satellite image with a corresponding timestamp and geolocation, the model predicts scene classes and visual attributes, which were annotated using off-the-shelf models (Zhou et al., 2017; Laffont et al., 2014), for matching ground-level images. The method relied on the conditional probability $P(a|t, l, I(l))$, where visual attributes $a$ are predicted based on timestamps $t$, geolocation $l$, and a satellite image $I(l)$. This approach assumes that visual attribute estimators model $P(a|I)$ accurately for ground-level images $I$. Therefore, it requires both geolocation $l$ and the accurate estimation of visual attribute $a$ to conversely model $P(t|I, a, l)$. This dependency limits its applicability to scenarios where the variables $l$ and $a$ are unavailable or inaccurate. In contrast, our work focuses on learning purely time-aware visual representations that directly connect natural images with their corresponding timestamps without such dependency.

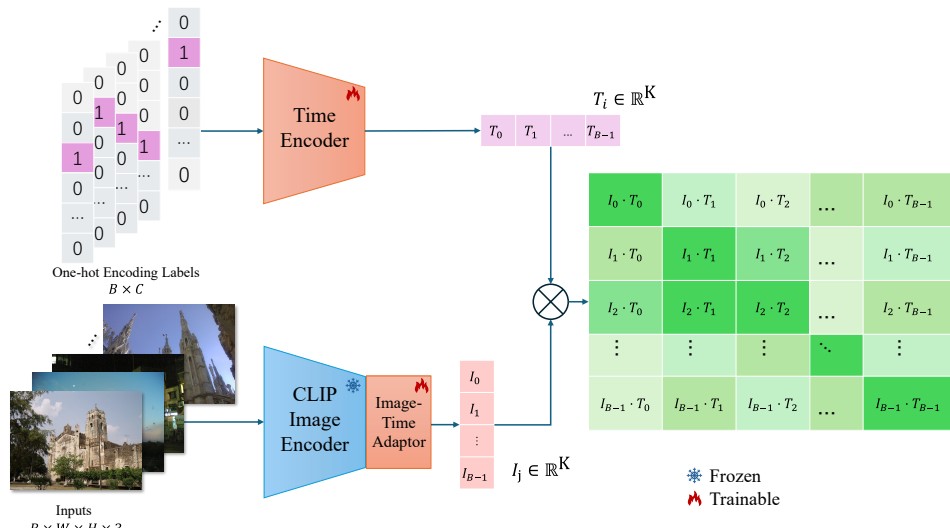

Figure 2: **An illustration of the proposed TICL pipeline.** Given the static images together with one-hot encoded time labels, two corresponding encoders (time encoder and image encoder + adaptor) are leveraged to project the input into feature vectors. Joint representation learning is applied on top of them in a contrastive learning manner.

## 3 METHODOLOGY

### 3.1 PROBLEM FORMULATION

Before introducing the proposed method, we revisit the problem formulation for timestamp estimation from images. In general, we seek to train a model $f_\theta(\cdot)$, predicts timestamp $\hat{t}$ given input images $x$. The estimation can be written as $\hat{t} = f_\theta(x)$. While regression seems ideal due to the continuous nature of time, it faces significant challenges (details in appendix A.2). The cyclic nature of the clock introduces discontinuity to regression methods treating target values as scalars within a range which is a disconnected set (Zhou et al., 2019). In regression, this issue to cyclic data often leads to estimations $\hat{t}$ around the midpoint of the whole range (Adams & Vamplew, 1998). For example, images with timestamps like 23:59 and 00:00, which are visually similar, appear at opposite ends of the time scale. In such case, the regression model tends to reach a sub-optimal solution that is around 12:00, which is far from accurate.

Most prior studies have employed classification over discrete time periods (*e.g.* hours), in which $\hat{t}$ has finite value options corresponding to classes. Classification mitigates the above issue in regression by assigning predictions to one of the discrete time classes. Even for boundary cases like 23:59 and 00:00, the classification model tends to predict one of the adjacent classes (*e.g.* 23:00 or 00:00), which is more reasonable. However, classification treats these classes as orthogonal one-hot vectors (Rodríguez et al., 2018), overlooking the relationships between time periods.

An effective time representation should capture the inherent relationships. With regard to this, we propose the Time-Image Contrastive Learning (TICL) method. TICL replaces one-hot encoding with learnable time embeddings that align with visual features. This approach can effectively capture inner relationships of time classes and enhances accuracy by leveraging contrastive learning between time and image representations.

### 3.2 TIME-IMAGE CONTRASTIVE LEARNING

We propose Time-Image Contrastive Learning (TICL), a multi-modal approach that jointly learns time and image representations using a cross-modal contrastive learning approach, inspired by Geo-CLIP (Vivanco et al., 2023). The general architecture of the model is illustrated in figure 2. Each input image $x_i$ is associated with a label $t_i \in \mathbb{R}^C$ indicating its time period. Empirically, we fix

$C = 24$ for all the results in the main paper for a fair comparison with previous works (further discussions on choice of $C$ in appendix A.3). Each one-hot encoded vector $t_i$ is projected into a high-dimensional representation space $\mathbb{R}^K$ using a Time Encoder $T_i = f_{\theta_T}(\cdot)$, where $K = 768$ to match the dimensionality of the image representation.

As illustrated in figure 2, during training iterations, input images are encoded by a frozen CLIP Image Encoder Radford et al. (2021). Following this, the Image-Time Adaptor module adjusts the CLIP image features to align with the time representations produced by the Time Encoder. The goal is to maximise the cosine similarity between the image feature $I_i = f_{\theta_I}(x_i)$ and its corresponding time-class embedding $T_i = f_{\theta_T}(t_i)$. Here, $f_{\theta_I}(\cdot)$ denotes the combined operation of the Image Encoder and the Image-Time Adaptor. The alignment is optimised by minimising contrastive loss function (He et al., 2019), as defined in equation 1, in which $\tau$ is a learnable temperature that controls the sharpness of the softmax distribution (Wu et al., 2018). As for inference, TICL flexibly supports classification and nearest-neighbour-based inference pipelines (see details in appendix A.4.3).

$$\mathcal{L}_B = -\sum_{i=0}^{B-1} \log \frac{\exp\left(I_i \cdot T_i / \tau\right)}{\sum_{j=0}^{B-1} \exp\left(I_i \cdot T_j / \tau\right)} \tag{1}$$

Several key intuitions support this design. Previous work has shown that combining additional geolocation and date information can improve the performance of time estimation. However, reliance on additional attributes may propagate errors from prior to posterior attributes (Salem et al., 2020). We observed that the CLIP image encoder is a powerful foundation model capable of capturing rich semantic contextual features from raw images (Agarwal et al., 2021). These context priors tend to incorporate effective cues (*e.g.* seasons, climate and regions) for timestamp estimation. Therefore, we use the frozen CLIP image encoder to directly extract these useful features. A standard MLP-based Image-Time Adaptor module is then applied to adapt the extracted CLIP features to the target time representations during training enabling their use for timestamp estimation tasks.

Another benefit of our design comes from the learnable time embedding in the contrastive learning scheme. In the vanilla classifier construction, the final output of the model $\hat{y}$ is constrained within the subspace of $\left\{||\hat{y}||_1 = 1, \hat{y} \in \mathbb{R}^C\right\}$, where each target label embedding is a fixed one-hot encoding mutually orthogonal to the others. Each image representation is optimised solely towards its own target, and thus the activation to other possible classes tends to be overwhelmed. This results in a significant bias towards majority classes when the dataset contains similar images with different classes, since the gradients towards the minority classes will be strongly offset by the majority classes with inputs that share similar features (He & Garcia, 2009). In contrast, our method provides each target time class with a trainable representation that is optimised to be aware of their corresponding sample prototypes, helping the model to align the representations of timestamps and visual inputs more effectively for tail classes (Zhu et al., 2022).

## 4 BENCHMARK DATASET TOC

In this paper, we introduce a new benchmark dataset – Time-Oriented Collection (TOC) – consisting of high-quality images sourced from social media, featuring calibrated and reliable image metadata. This dataset reflects real-world scenarios and human activities, improving the applicability of time of day estimation in practical tasks. We collected 117,815 training samples and 13,091 test samples from the CVT dataset, mitigating various limitations in previous datasets.

Previous datasets, such as YFCC100M (Thomee et al., 2016) and Cross-View Time (CVT) (Salem et al., 2020), contain unnatural non-photographic images (*e.g.* memes, scribbles) and inaccurate timestamps due to unsynchronised clocks and other sources of inconsistency. Furthermore, stationary webcam-based datasets lacked sufficient diversity to represent random natural views at different times of the day. To mitigate these issues, we implemented a two-step filtering and calibration process for the CVT dataset. Specifically, we applied DBSCAN clustering (Ester et al., 1996) to PCA-reduced features extracted from a pretrained ResNet-18 model (He et al., 2015) to eliminate unnatural images marked as outliers. A subsequent manual review was conducted to further remove images with incorrect timestamps or poor quality. This updated dataset reflects natural variations in human activity throughout the day with improved reliability in terms of time metadata (more details in appendix A.1.1).

Table 1: Time estimation performance on our TOC dataset and the AMOS test set.

| | **TOC test set** | | | | **AMOS test set**[†] | | | |
|---|---|---|---|---|---|---|---|---|
| | Top-1 acc ↑ | Top-3 acc ↑ | Top-5 acc ↑ | Time MAE (min.) ↓ | Top-1 acc | Top-3 acc | Top-5 acc | Time MAE (min.) |
| SVM (Sharma et al., 2016) | 1.80% | 6.27% | 12.06% | 450.18 | 1.55% | 6.66% | 14.14% | 435.86 |
| ResNet-101 (Zhang et al., 2022) | 13.24% | 37.30% | 58.23% | 177.84 | 7.85% | 24.26% | 40.10% | 261.89 |
| ViT-B/32 (Zhang et al., 2022) | 10.73% | 31.21% | 49.05% | 195.33 | 7.25% | 21.03% | 32.93% | 263.87 |
| Zhai et al. (2019) | 14.11% | 40.47% | 65.94% | 188.78 | 9.14% | 27.95% | 45.36% | 262.68 |
| Salem et al. (2022) | 13.87% | 39.36% | 60.71% | 186.44 | 8.63% | 26.49% | 42.58% | 255.20 |
| **TICL (Ours)** | 20.60% | 49.01% | **67.82%** | 171.65 | **13.55%** | **38.49%** | **57.28%** | **187.87** |
| **TICL-Nearest-Neighbour (Ours)**[‡] | **25.67%** | **49.32%** | 66.74% | **156.24** | 11.14% | 31.01% | 48.84% | 220.94 |
| Zhai et al. (2019)[§] | 15.01% | 42.54% | 68.24% | 185.34 | 8.85% | 24.12% | 38.63% | 268.41 |
| Salem et al. (2022)[§] | 13.53% | 38.47% | 59.10% | 176.70 | | | | 257.00 |

[†] Experiments on this test set are conducted under a zero-shot manner, in which we directly evaluate models trained solely on TOC dataset.
[‡] Results in this row are achieved via Nearest-Neighbour style inference. We directly choose the timestamp labels of nearest neighbours in terms of representations from train dataset as estimations (see appendix A.4.3 for more details).
[§] These methods take additional known geolocation metadata inputs. Therefore, it's unfair to directly compare them with other methods. So we put them here just for reference.

## 5 EXPERIMENTS

### 5.1 TIME ESTIMATION PERFORMANCE

We use different evaluation metrics to measure performance on timestamp estimation tasks: top-k classification accuracy with $k = 1, 3, 5$, and Time Mean Absolute Error (MAE) on a minute basis. In addition to the TOC test set, to better evaluate the generalisation ability of the proposed method, we selected a subset of the AMOS dataset (Jacobs et al., 2007) as an additional test set. This additional test set contains 3,556 high-quality images captured by several stationary surveillance cameras with a more balanced timestamp label distribution. That is, our model is trained solely on the TOC training set and evaluated on different test sets to demonstrate generalisation ability across different domains (experiment hyper-parameters in appendix A.4.1).

As shown in table 1, TICL not only outperforms all previous pure vision methods but also outperforms previous methods that require additional geolocation inputs on most metrics. TICL also demonstrates better performance in the additional AMOS test set, thereby indicating better generalisation ability.

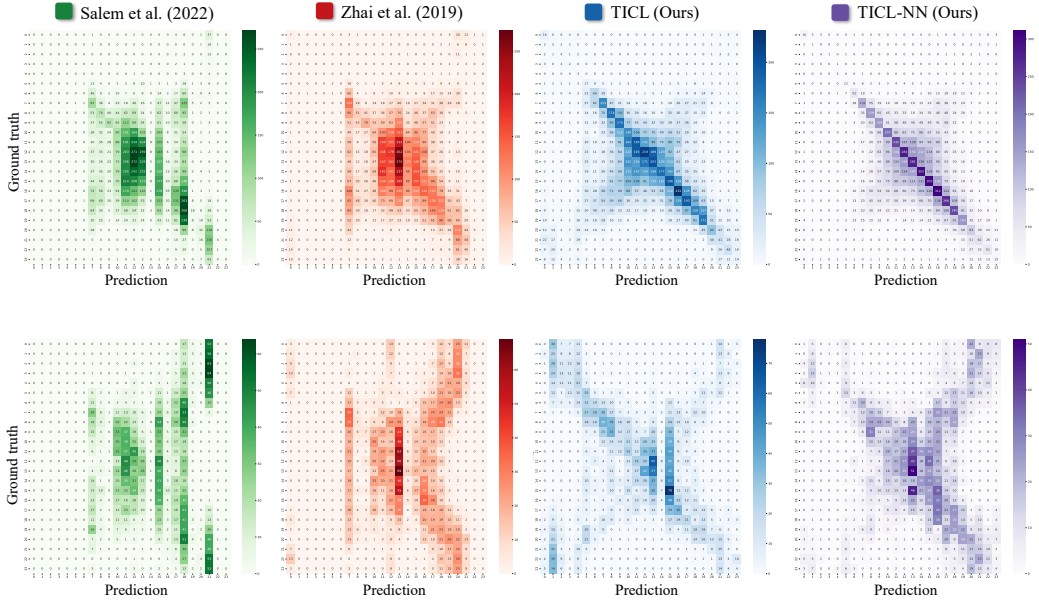

Figure 3: **Confusion matrices.** They provide more detailed comparisons throughout the 24 hours on our TOC test set (top), and the AMOS test set (bottom).

Apart from the quantitative results of class accuracy and time MAE, we also visualised the confusion matrices in figure 3 to provide a more in-depth evaluation of the task. An interesting finding is that both Salem et al. (2022) and Zhai et al. (2019) overlooked minority classes in the training set (classes

Table 2: Ablation study of the proposed method design.

| Image Encoder[†] | $f_{\theta_T}$[‡] | $f_{\theta_{ITA}}$[§] | TOC test set | | | | AMOS test set | | | |
|---|---|---|---|---|---|---|---|---|---|---|
| | | | Top-1 acc ↑ | Top-3 acc ↑ | Top-5 acc ↑ | Time MAE (min.) ↓ | Top-1 acc | Top-3 acc | Top-5 acc | Time MAE (min.) |
| EfficientNetV2(L) | ✗ | ✗[¶] | 6.93% | 21.59% | 35.09% | 299.17 | 6.18% | 20.13% | 33.52% | 294.68 |
| | ✓ | ✗ | 6.92% | 20.82% | 34.96% | 303.51 | 6.21% | 20.44% | 34.95% | 291.71 |
| | ✗ | ✓ | 7.56% | 23.04% | 37.59% | 276.40 | 6.83% | 20.61% | 33.41% | 280.35 |
| | ✓ | ✓ | 8.52% | 23.71% | 38.63% | 258.75 | 7.54% | 21.57% | 35.34% | 277.85 |
| DINOv2-base | ✗ | ✗ | 7.69% | 23.36% | 38.61% | 302.84 | 5.65% | 17.12% | 27.28% | 319.09 |
| | ✓ | ✗ | 8.01% | 23.84% | 39.06% | 295.34 | 5.23% | 17.35% | 29.22% | 320.76 |
| | ✗ | ✓ | 1.02% | 3.29% | 12.04% | 486.77 | 4.11% | 11.41% | 19.62% | 381.92 |
| | ✓ | ✓ | 9.53% | 27.34% | 44.17% | 254.49 | 5.09% | 14.74% | 25.16% | 327.72 |
| SwinV2(B) | ✗ | ✗ | 11.45% | 32.27% | 51.08% | 240.77 | 7.87% | 22.49% | 36.81% | 281.80 |
| | ✓ | ✗ | 11.64% | 32.13% | 50.33% | 243.86 | 7.51% | 22.36% | 37.54% | 288.21 |
| | ✗ | ✓ | 12.74% | 33.65% | 52.06% | 222.76 | 6.75% | 23.76% | 38.41% | 284.30 |
| | ✓ | ✓ | 13.37% | 34.94% | 52.93% | 216.17 | 7.37% | 22.98% | 38.08% | 276.66 |
| ConvNeXt(L) | ✗ | ✗ | 11.59% | 32.93% | 50.88% | 240.64 | 6.41% | 21.68% | 37.63% | 300.74 |
| | ✓ | ✗ | 11.86% | 32.81% | 50.18% | 240.80 | 6.10% | 20.66% | 35.85% | 302.45 |
| | ✗ | ✓ | 13.51% | 35.29% | 52.76% | 216.28 | 7.71% | 24.33% | 39.96% | 275.23 |
| | ✓ | ✓ | 14.67% | 36.75% | 54.60% | 204.19 | 8.27% | 24.78% | 40.86% | 263.03 |
| **CLIP (ViT-L/14)** | ✗ | ✗ | 16.66% | 44.43% | 65.07% | 193.66 | 12.37% | 36.95% | 55.96% | 200.93 |
| | ✓ | ✗ | 16.73% | 44.05% | 63.99% | 195.41 | 13.50% | 38.49% | **58.30%** | 189.99 |
| | ✗ | ✓ | 18.60% | 46.41% | 65.98% | 181.22 | 12.57% | 37.51% | 57.23% | 189.69 |
| | ✓ | ✓ | **20.61%** | **49.01%** | **67.83%** | **171.65** | **13.55%** | **38.50%** | 57.28% | **187.87** |

[†] All image encoders are frozen feature extractors with pretrained features provided by corresponding PyTorch libraries (Wolf et al., 2020).
[‡] $f_{\theta_T}$ denotes the Time Encoder module. When $f_{\theta_T}$ is absent, only one-hot encoding is used to represent the timestamp, and the outputs of $f_{\theta_I}$ need to be projected to 24 dimensions to match the timestamp encoding.
[§] $f_{\theta_{ITA}}$ denotes the Image-Time Adaptor. When it is absent, only the backbone feature extractor and time encoder are used.
[¶] The baseline with neither of the $f_{\theta_T}$, $f_{\theta_{ITA}}$ components simply has a linear layer after Image Encoder projecting the features to 24 dimensions.

from 1 a.m. to 5 a.m.), resulting in nearly no predictions for these classes on both test sets. This indicates a notable bias in these models towards classes during hours of intense human activity, when more images are present in the datasets. In contrast, our proposed TICL method exhibit more balanced distributions of positive predictions between classes on two test sets, suggesting better prediction fairness.

The general trend in all the confusion matrices also suggests the remaining challenges faced by all methods. Notable anti-diagonal patterns indicate inherent visual ambiguities of the clock system. Addressing these ambiguities requires the incorporation of priors that are not directly available in standard vision models. For instance, distinguishing between images captured during sunset or sunrise is relatively straightforward when the view orientation and location are known. However, for purely vision models lacking awareness of these clues, such scenarios are very challenging. Additionally, nearly all models struggle to classify adjacent classes, which is natural, since samples within these classes share similar lighting conditions, making the differences too subtle to discern.

In summary, our experimental results indicate an overall improvement of the proposed methods in timestamp estimation, especially in terms of accuracy, class-wise prediction fairness, and generalisation ability.

## 5.2 ABLATION STUDY

In this section, we present the ablation study, evaluating each module of the proposed TICL model across different configurations. To ensure a fair comparison, we used a classification-based inference pipeline for all experiments (details in appendix A.4). Table 2 provides performance comparisons under various settings, including different backbones (Tan & Le, 2021; Oquab et al., 2023; Liu et al., 2022a;b) within the image encoders.

The differences in performance across the image encoder backbones highlight the effectiveness of the CLIP Image Encoder. Thanks to its rich semantic representations, the CLIP Image Encoder consistently achieves better results across all configurations than other backbones. Additionally, we observed that the Time Encoder $f_{\theta_T}$ and the Image-Time Adaptor $f_{\theta_{ITA}}$ have varying effects when used individually, either slightly improving or degrading the baseline. However, when these two modules are employed simultaneously, they lead to universal improvements across all image encoder backbones. This underscores the importance of the joint contribution of the Time Encoder and Image-Time Adaptor in effectively aligning time and image representations.

### 5.3 Investigation on downstream tasks

To study the effectiveness of the learned time-aware representations, in this section, we explore several time-related downstream tasks under a zero-shot setting. Specifically, we directly use the TICL model trained on TOC without any further fine-tuning. This allows us to directly inspect the capabilities of the learned time-aware representation across multiple aspects.

#### 5.3.1 Time-based image retrieval

An intuitive application of time-aware representations is the time-based image retrieval. It aims to effectively retrieve images from a database with a similar captured time of day to the query images. We designed a zero-shot vector search engine that retrieves the nearest neighbours of query images based on their time-aware representation similarities. To evaluate the retrieval task, we separated the TOC test set into 13,043 database images and 48 query images spanning all 24 hours. The performance is measured using Recall@k (Arandjelović et al., 2016) reported in figure 4. Images retrieved with a time difference of no more than 30 minutes from the queries are considered as positives. The results clearly show that the proposed TICL model achieves the best performance across all Recall@k metrics.

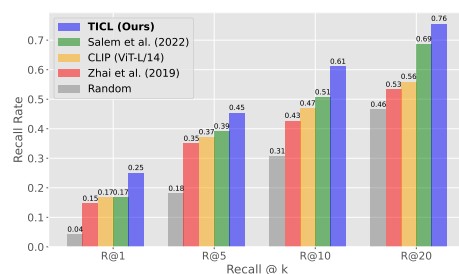

Figure 4: Performance (Recall@k) comparison on time-based image retrieval.

Another interesting finding from the results is that representations from other time-estimation models Salem et al. (2022); Zhai et al. (2019) occasionally outperform CLIP representations. A possible explanation is that they modelled the conditional probability of metadata including both dates and timestamps. Such joint prediction allows the model to memorise the composition of metadata w.r.t. visual inputs. However, such a composition does not always generalise to reality, where the correlation between these attributes is not always deterministic (Ghosh et al., 2017). In contrast, the time representations learned from solely visual inputs by our proposed TICL model already outperforms all other representations without utilising any metadata other than time (more results and analysis in appendix A.6).

#### 5.3.2 Video scene classification

Understanding dynamic scenes is an important challenge that deep learning models currently face (Miao et al., 2021). A fundamental task in this domain is the video scene classification. To assess whether our time-aware representations provide valuable priors for understanding different categories of dynamic scenes, we designed a model architecture that concatenates the VideoMAE (ViT-B) backbone (Tong et al., 2022) with a frozen TICL feature extractor, followed by a linear classification head. This allows us to test the contribution of incorporating time-aware representations into this task. We test on various scene datasets including Hollywood2-Scene (Marszałek et al., 2009), YUP++ (Derpanis et al., 2012) and 360+x (Chen et al., 2024) (details and discussion in appendices A.5 and A.7).

We compared the performance under different additional feature extractors. According to table 3, TICL representations provide a substantial improvement to the scene classification task. The most straightforward explanation for this boost is that scene classes are correlated with learned time-of-day representations. For example, "Breakfast" often happens during the morning (see appendix A.9). In addition, as we have shown in section 5.3.1, the TICL representations can capture similarities between images with close timestamps. Natural videos, although they sometimes involve drastic subject or view movement, frames within each should still represent a continuous time periods. TICL representations for frames across the whole video should be more consistent than those of vanilla CLIP, which have stronger locality per frame (Tang et al., 2021). This intra-video consistency allows for more general time-aware priors extracted using TICL. The t-SNE visualisation of the video features in figure 5 supports this claim, showing that TICL features are more separable than

Table 3: Performance comparisons on the video scene classification task.

|  | Hollywood2-Scene ↑ | YUP++[†]↑ | 360+x (Panoramic) ↑ | 360+x (Third-person) ↑ |
|---|---|---|---|---|
| Vanilla VideoMAE | 18.73% | 97.29% | 53.70% | 54.55% |
| VideoMAE + CLIP (ViT-L/14) | 22.51% | **98.33%** | 57.40% | 50.91% |
| VideoMAE + Salem et al. (2022) | 32.99% | 97.50% | 44.45% | 52.72% |
| VideoMAE + Zhai et al. (2019) | 32.65% | 97.71% | 48.15% | 56.36% |
| **VideoMAE + TICL (Ours)** | **59.79%** | **98.33%** | **59.26%** | **58.18%** |

[†] We use an unofficial train/val/test split of 5:1:4, since original 1:9 train/test split overfit prematurely.

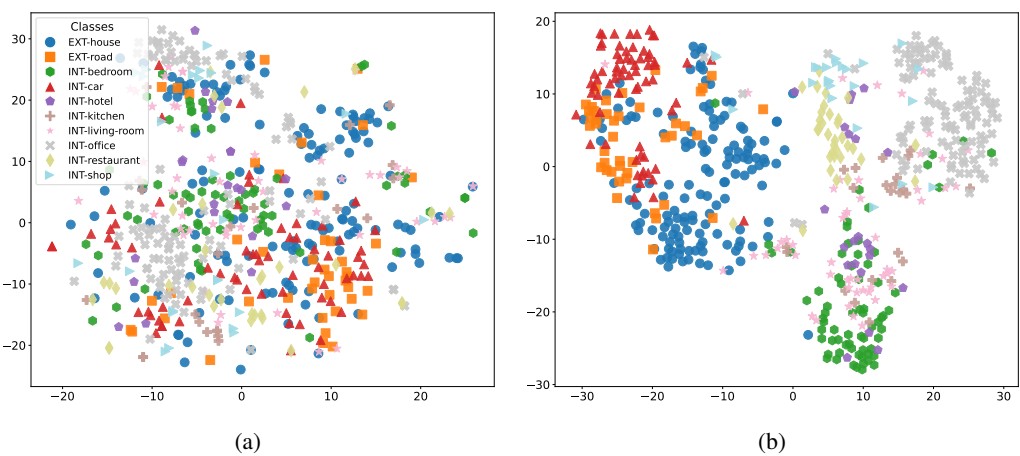

(a)             (b)

Figure 5: **t-SNE visualisation comparison.** It compares time-aware video representations learned from (a) CLIP and (b) our TICL, in the Hollywood2-Scene dataset (Marszałek et al., 2009).

vanilla CLIP features (see appendices A.7 and A.9 for a more in-depth analysis of the phenomena and claims above).

### 5.3.3 TIME-AWARE IMAGE EDITING

As aforementioned in section 3.2, the TICL model provides the corresponding embeddings for certain periods of the day. Therefore, it is natural to consider using these timestamp embeddings as guidance to edit images toward different periods to examine this conjecture. To assess the extent to which timestamp embeddings aid this task, we adopted the following experiment framework from Patashnik et al. (2021) that conducts image editing via latent vector searching through optimisation steps instead of tuning the models.

To provide comprehensive evaluations, we conducted experiments on three different baseline Style-GAN2 models (Karras et al., 2020b) focusing on different subjects trained on (Skorokhodov et al., 2021; Yu et al., 2015). The pretrained weights are adopted from the codebases by Pinkney (2024); Epstein et al. (2022); Karras et al. (2020a) respectively. The editing pipelines were restricted to follow the same latent optimisation baseline method introduced in StyleCLIP (Patashnik et al., 2021). Additionally, we designed a new timestamp-aware synergy loss combining directional CLIP loss and TICL loss (implementation details in appendix A.5).

The proposed timestamp-aware synergy loss yields the most plausible synthesis outcome as illustrated in figure 6. The limitations of solely text-guided image editing methods could be due to their susceptibility to certain adversarial solutions fooling CLIP text representations with certain patterns only (Liu et al., 2021). Specifically, figure 6 shows the vanilla StyleCLIP edits using the CLIP loss tend to focus on the general tint of the image but fail to reflect realistic illuminations. We find that replacing the CLIP loss with a directional variant introduced in previous works (Gal et al., 2021; Kwon & Ye, 2022) can assist in overcoming larger domain gaps. Despite showing improvements over the baseline editing method, the results still show unrealistic artefacts and shape distortions. These limitations show the necessity of incorporating additional time representations we proposed other than just text representations when computing loss functions for image edits. Our qualitative evaluations demonstrated the effectiveness of the TICL representations on the specific task. We also

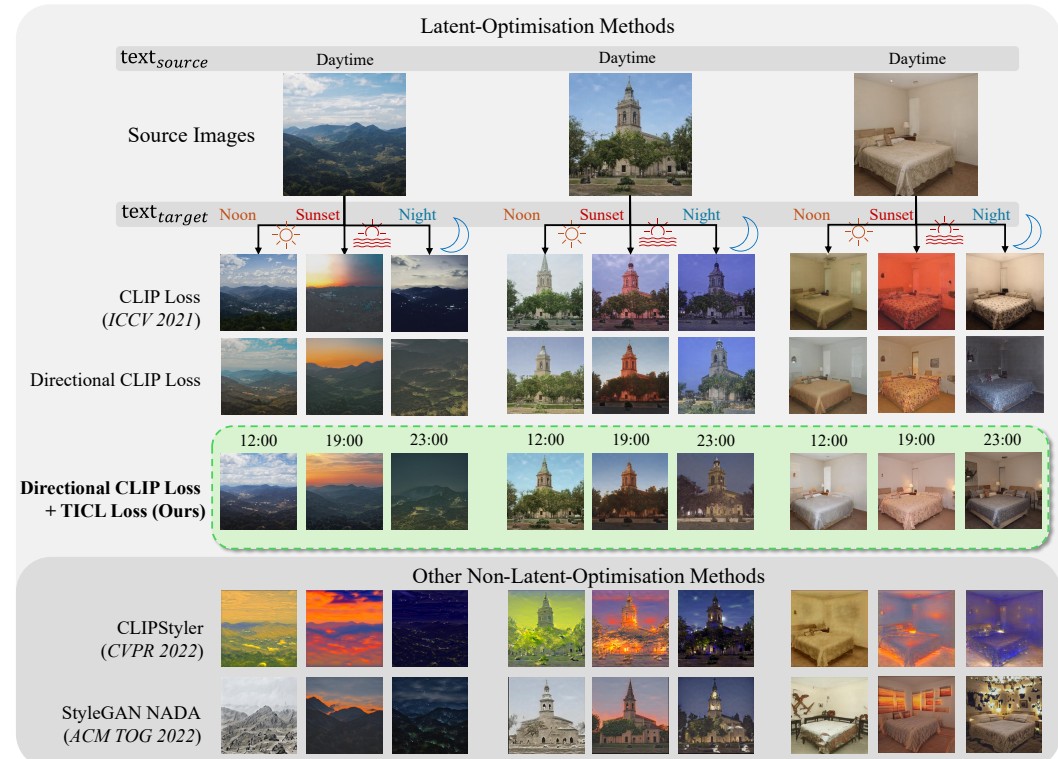

Figure 6: **Time-aware image editing.** It shows the results of applying our timestamp-aware editing method (green overlay) on three different StyleGAN2 models trained on LHQ-Landscape (Skorokhodov et al., 2021), LSUN-Church, and LSUN-Bedroom (Yu et al., 2015) datasets. The results of other non-latent optimisation methods are also demonstrated (grey overlay).

included other previous baseline method results that work under different frameworks other than latent optimisation for a more comprehensive comparison (more details and quantitative results in appendix A.8).

In conclusion, despite the inherent limitations of the latent optimisation-based editing methods, which struggle with editing images to unseen domains for generators (Patashnik et al., 2021), the comparisons of different editing results still provide interesting observations regarding the capability of time-aware representations in aiding visual generative tasks.

## 6 CONCLUSION

In this paper, we tried to answer the question of *what time tells us*, by looking at the time-aware representations learned from static images, through the task of time-of-day estimation. A new reliable benchmark dataset, *TOC* was introduced to support the task, consisting of images captured in natural settings with verified timestamps. This dataset addresses the limitations of existing datasets by providing a more diverse and realistic collection of images that better reflect daily visual experiences. Build upon that, a new learning paradigm (*TICL*) was proposed, which jointly models timestamp and image representations via cross-modal contrastive learning, surpassing previous works in time-of-day estimation. The learned time-aware representations were further studied by validating on several downstream tasks. The strong performance in these downstream tasks highlighted its capability to recognise the similarity of the captured time (in time-based image retrieval), TICL's frame-coherent priors for video scene understanding (significantly improved video scene classification), and realistic and time-consistent performance in time-aware image editing (accurately reflecting typical lighting conditions for different times of day). In summary, by answering the question at the beginning, our study showed the potential of learning time-aware visual representations from static images, and its benefits to various time/temporal-related downstream tasks, suggesting the visual essence of time.

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

# A APPENDIX / SUPPLEMENTAL MATERIAL

## A.1 MORE DETAILS ON DATASETS

### A.1.1 THE PROPOSED TOC DATASET

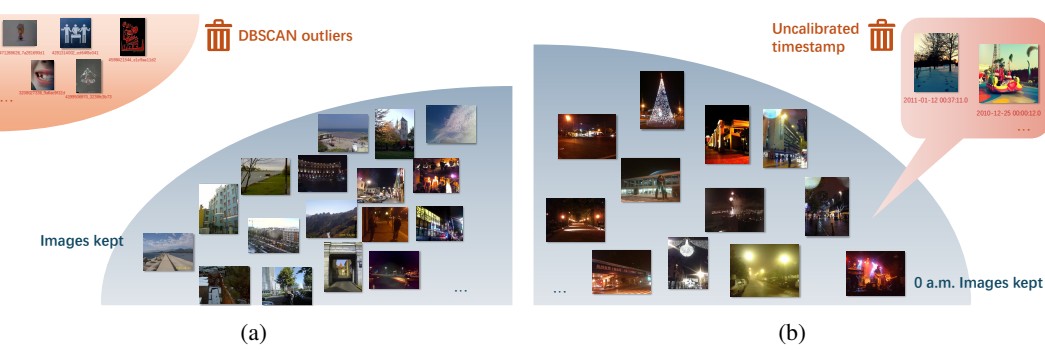

(a)                                                        (b)

Figure 7: **Dataset filtering process,** where (a) shows how DBSCAN (Ester et al., 1996) removes unnatural images that may degrade dataset quality, and (b) shows examples of removed images with uncalibrated timestamps.

In this work, we introduce a new benchmark dataset in section 4 that combines images from the YFCC100M (Thomee et al., 2016) and Cross-View Time datasets (Salem et al., 2020). This section covers more details of the dataset. Figure 7 gives a clear illustration of the data filtering steps to the dataset, improving the sample quality and metadata reliability.

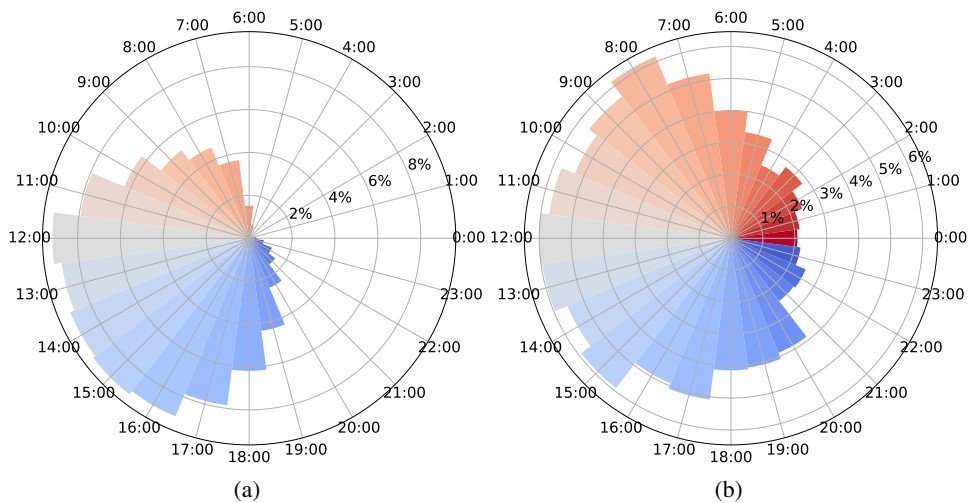

(a)                                                        (b)

Figure 8: **Dataset hourly sample distribution,** where (a) shows hourly sample distribution for TOC dataset, in which daytime images are significantly more prevalent than nighttime images, and (b) shows hourly sample distribution for AMOS-test dataset displaying a similar skewed but more balanced distribution towards daylight hours.

Following the data filtering, we partitioned the TOC dataset into a training set and a test set at a 9 : 1 ratio, with stratified sampling to ensure that the time distributions of both subsets were approximately equivalent. We observed a significant scarcity of images with reliable metadata captured at night compared to daytime images. This observation corroborates our hypothesis that the distribution of timestamps in images shared on social media is inherently unbalanced as depicted in figure 8. Such imbalance presents challenges in learning equitable representations for time periods that are underrepresented due to limited sample availability. This imbalance necessitates strategic approaches to model training that can adequately compensate for these discrepancies.

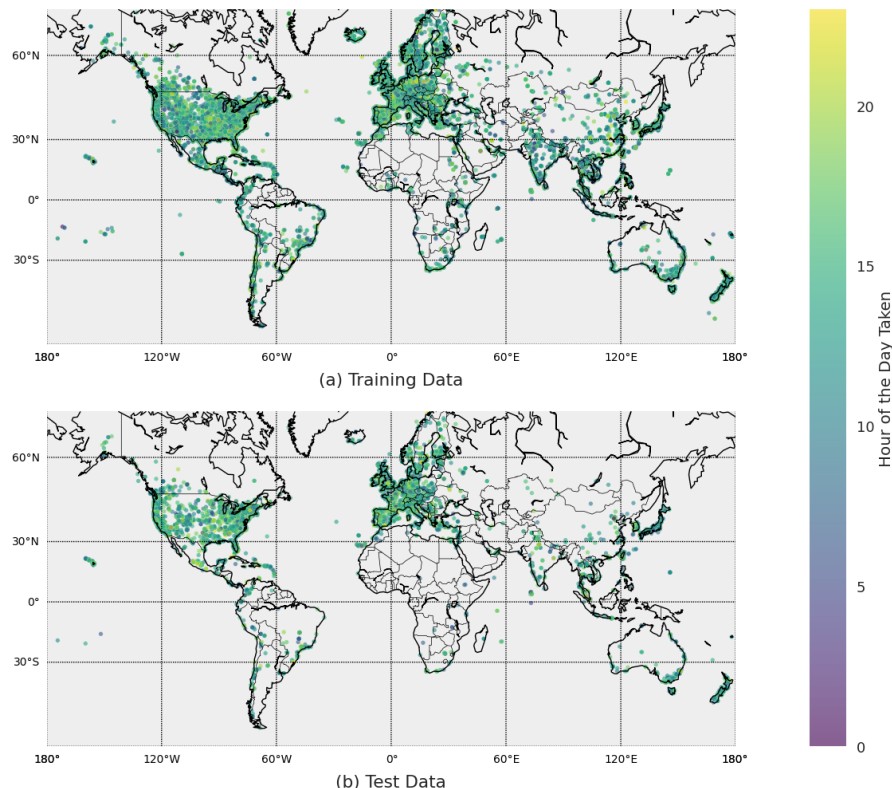

Figure 9: **Geo-temporal distribution of TOC.** Each point corresponds with a sample located on the map and the color represents its captured time of day.

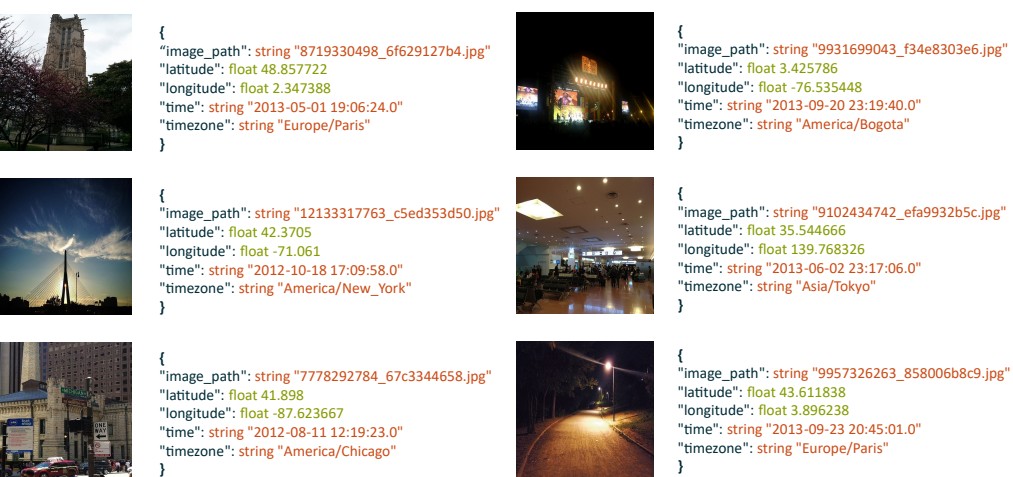

Figure 10: **Sample images and metadata from the TOC dataset.** Metadata contains several fields indicating timestamps and geolocations.

We present the following visualisations for the image diversity in the dataset. Figure 9 visualises the geolocation and temporal distribution of both train/test datasets, showing the broad temporal and spatial coverage of the images. Figure 10 provides a few examples of the exact format and appearances of the samples within the TOC dataset, in which each image sample is paired with their corresponding metadata.

### A.1.2 AMOS TEST DATASET

The AMOS test dataset is derived from the Archive of Many Outdoor Scenes (AMOS) dataset (Jacobs et al., 2007), containing 3,556 images captured by 53 stationary surveillance cameras. The dataset construction involves several steps to ensure metadata reliability and sample quality. First, we calibrated the original UTC timestamps to their respective local time zones using the geolocation metadata. Then, we filtered out 1) noisy images with low Peak Signal-to-Noise Ratio (PSNR) and 2) underexposed images with low average and standard deviation for pixel brightness, ensuring that only high-quality images were included. Figure 11 shows a few sample images from the dataset.

As the images were captured automatically by surveillance cameras with fixed views, the AMOS test set represents a different domain to the proposed TOC dataset. Although the dataset contains repetitive visual appearances due to the stationary setup of the cameras, it benefits from a more balanced distribution of timestamps throughout the day, as shown in figure 8b.

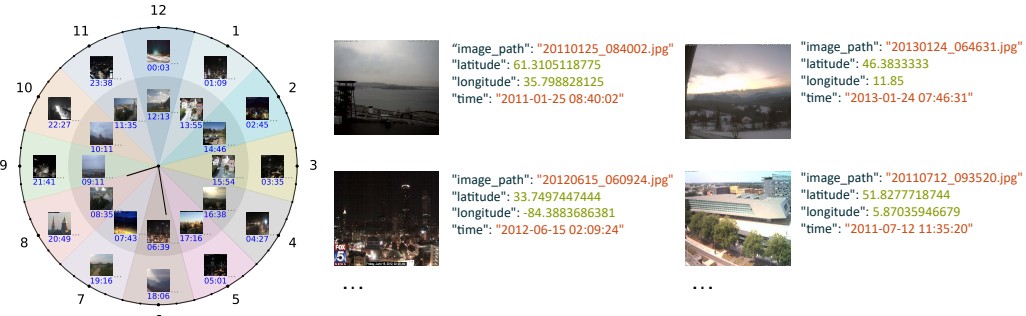

Figure 11: **Sample images from the AMOS test dataset.** The images showcase different scenes captured by stationary surveillance cameras at various times of the day.

### A.2 LIMITATIONS OF REGRESSION MODEL

The regression style construction for timestamp estimation presents significant challenges as covered in section 3.1. There are different issues with regression models, including 1) loss function sensitivity and 2) discontinuity in the scalar range for regression. In the following paragraphs, we first provide a brief illustration of the issue on the regression loss function. Secondly, we present experiments of a regression model working in a circular space instead of the vanilla scalar range which is a disconnected set. These experiments provide possible explanations for the limits of vanilla regression models.

Let us define the problem setting of timestamp regression as follows. Given an image $x$, the objective is to predict the timestamp $y$ in the range $[0, 24)$ hours of the day. In a regression framework, the model $f_\theta$ maps an input image $x$ to a continuous scalar output $\hat{y} = f_\theta(x) \in [0, 24)$.

Consider a dataset $\mathcal{D}$ consisting of images taken at various times throughout the day. Specifically, consider pairs of images $\{(x_i, y_i), (x_j, y_j)\}$ taken during "symmetric times" such as sunrise and sunset, where the general light conditions are similar but the ground truth timestamps are different (see figure 12d and 12c). With very similar inputs and the same model $f_\theta(\cdot)$, it holds that:

$$f_\theta(x_i) \approx f_\theta(x_j)$$

Then the Mean Squared Error (MSE) loss for the regression model over the dataset is defined as:

$$\mathcal{L}_{\text{MSE}}(\theta) = \frac{1}{|\mathcal{D}|} \sum_{k=0}^{|\mathcal{D}|-1} (y_k - f_\theta(x_k))^2$$

To find the optimal model parameters $\theta^*$, we minimise this loss function, ideally, the optimiser should reach:

$$\nabla_\theta \mathcal{L}(\theta) = 0$$

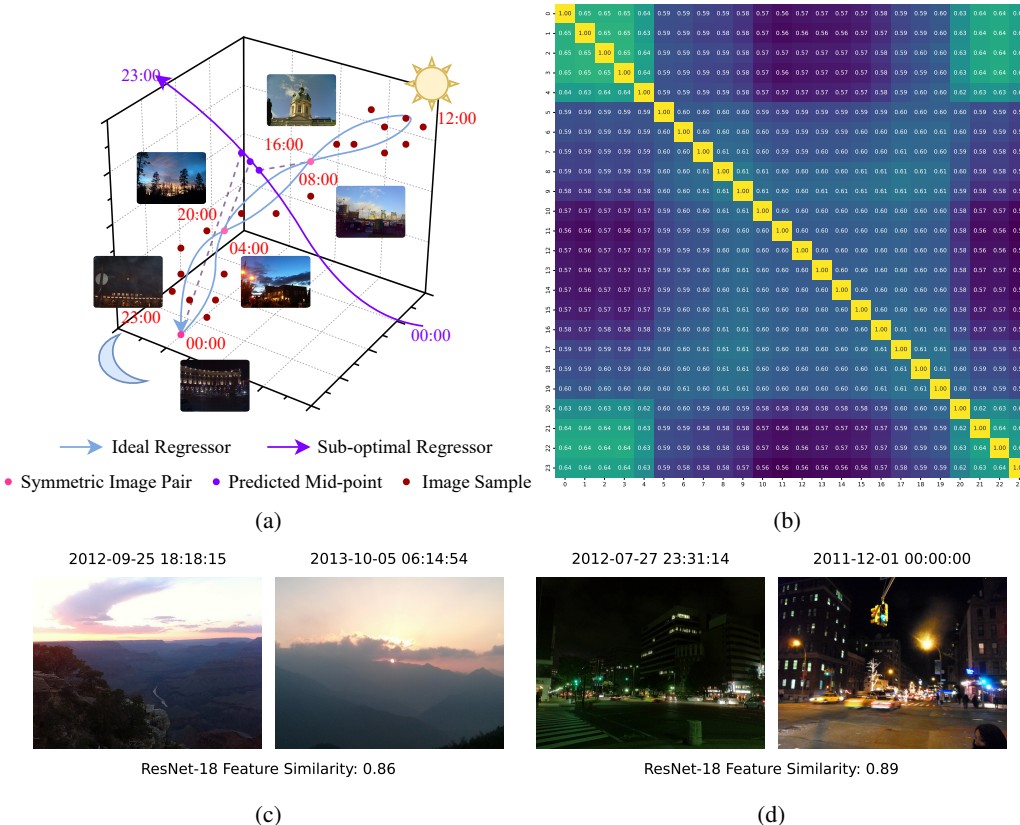

Figure 12: **Visual ambiguities for ground truth in regression.** (a) depicts a sub-optimal regression model where the predictions are biased towards the mid-point, and (b) shows a trend that images with more similar ResNet-18 features could have disparate timestamps. Few examples of such cases are provided in (c), (d).

For pairs of similar images with different $y$, this optimisation leads to mid-point predictions:

$$\hat{y}_i \approx \hat{y}_j \approx \frac{y_i + y_j}{2}$$

This effect leads to local minima in the timestamp representation space in figure 12a, particularly when $y_i$ and $y_j$ are at opposite ends of the 24-hour cycle, for example, 00:00 and 23:59. The regression model struggles with the ambiguous nature of time, resulting in systematically biased predictions towards the midpoint of symmetric times. Such bias results in incorrect gradient updates that cannot lead to an accurate timestamp estimation model for inputs $x_i, x_j$.

The aforementioned phenomenon of similar images with disparate ground truth timestamps prevails in the dataset. As evidence, we visualise the similarity of the ResNet-18 feature throughout hours for the entire dataset in figure 12b. Therefore, this overall trend of feature similarity extends the reasoning to the entire dataset, where the predictions $\hat{y}$ are systematically biased towards the average of the timestamp distribution. The prediction is likely to follow the normal distribution with the same mean value to the ground truth distribution and smaller variance $\sigma$ (Murphy, 2012).

$$\hat{y} \sim \mathcal{N}\left(\frac{1}{|\mathcal{D}|}\sum_{k=0}^{|\mathcal{D}|-1} y_k, \sigma\right)$$

We conduct corresponding experiments to provide evidence for the claims above. Particularly, we train a regression model using ResNet-101 backbone. The prediction histogram and confusion matrix provided in figure 13a and figure 13b support our claims. The predictions are heavily concentrated around the average value of the ground truth distribution, while the actual timestamps in the

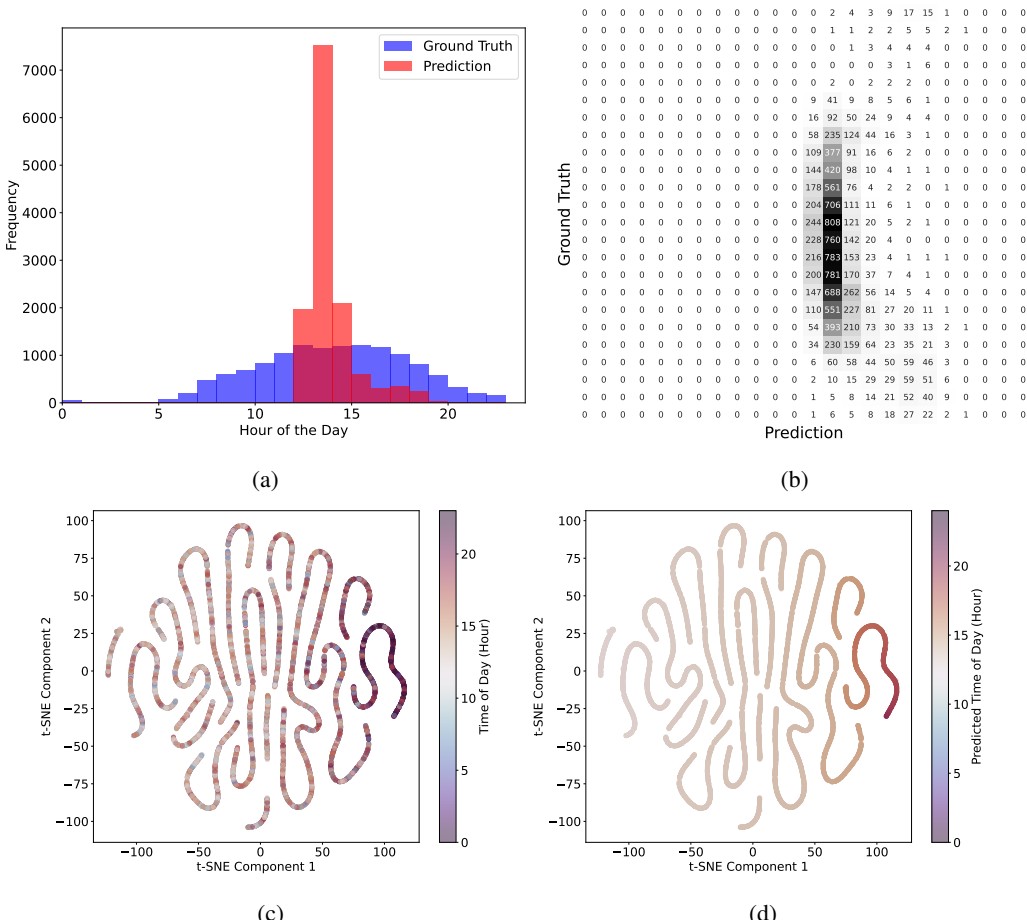

Figure 13: **Experiments on regression model.** (a) shows prediction distribution of regression model on TOC test set, (b) represents the confusion matrix by hour, (c) and (d) visualise t-SNE of regressor representations annotated with ground truth and predicted timestamps, respectively.

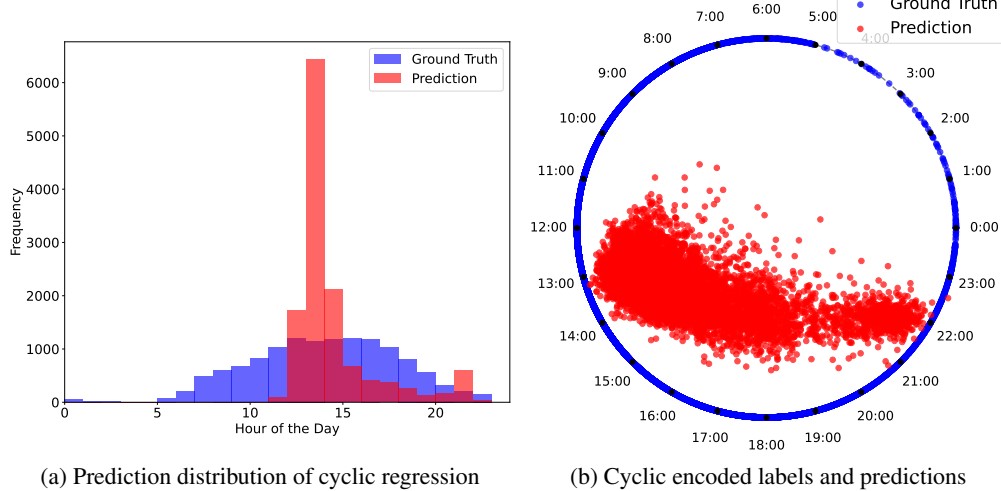

(a) Prediction distribution of cyclic regression  (b) Cyclic encoded labels and predictions

Figure 14: **Cyclic regression model results,** (a) shows prediction distribution of cyclic regression model, and (b) visualise how the cyclic encoding of predictions differ from the ground truth.

dataset are more evenly distributed throughout the day. This discrepancy highlights the failure of the regression model to capture the cyclic nature of time, resulting in biased predictions of the average of the whole range. Figure 13c shows that the regression model fails to discern similar images with different timestamps, where the representation forms disjoint trails on which representations of images from totally different time periods are nearly overlapped with each other. Figure 13d further shows how the regression model predicts average timestamps for these images with overlapping representations. These phenomena show that although the regression model managed to learn a certain extent of continuity of time of day from static views, it failed to tackle the ambiguity of timestamp given visual inputs with similar illuminations. Therefore, while such a regression model reaches convergence at local minima for the MSE loss, it is not ideal resorts we are looking for.

As we identified in section 3.1, the regression range is a disconnected set. Here we present an attempt to solve the discontinuity of the timestamp scalar range, we adopted a previous method bridging the gap by trigonometric encoding and decoding to cyclic data (Adams & Vamplew, 1998). Specifically, it encodes the scalar data $y$ into points on the unit circle $(\cos{(y/y_{max})}, \sin{(y/y_{max})})$, and decodes the model outputs in the reverse direction. Such representation space is proved to be continuous (Zhou et al., 2019). It bridged the gap between the end and the start of the regression value range, which was supposed to be close. We tried this remedy and found that it slightly mitigates the issue of over-concentration on the average values, as shown in figure 14a.

However, although this modification managed to rescue part of night images that are wrongly predicted toward the mean value of the whole target value range, it still exhibits poor prediction fairness, with most of the predictions falling in certain short time spans. The possible cause for such phenomena could still be the local minima that persist in the MSE loss landscape due to the prevailing timestamp ambiguities we discussed. Another observation in figure 14b is that there exists an obvious gap between the distribution of trigonometric encoding of ground truth timestamps and the predictions. This suggests that the cyclical correlation between visual representations and clock time may not perfectly follow the simple unit-circle assumption in Adams & Vamplew (1998). In contrast, our proposed learnable representations for the target labels in TICL can capture more complex correlations between different periods of clock times and visual cues without imposing such assumptions.

Therefore, the regression approaches to timestamp estimation struggle to properly address the ambiguity between time and visual features. Regression-based solutions are thus not as favourable for the pretext task of timestamp estimation.

### A.3 ABLATION STUDY ON CLASS PARTITIONING

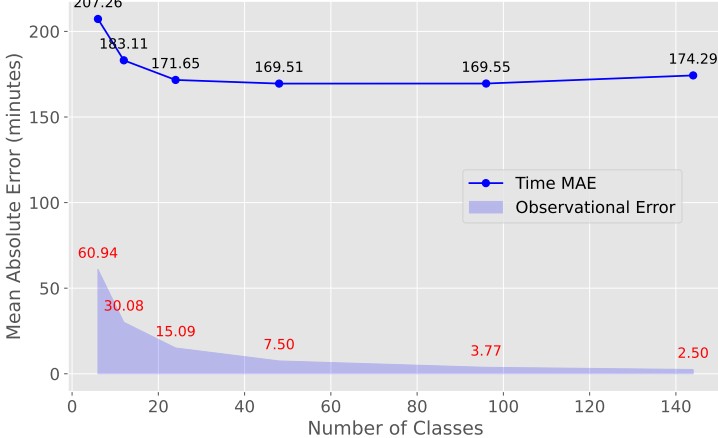

Figure 15: **Comparative error analysis of different class partitioning schemes,** it shows trends of mean absolute error (MAE) and observational error.

In the main paper, we adhere to the 24-class classification scheme used in previous methods, as stated in section 3.2. In this section, we explore the effects of different granularities of class partitioning. Using discrete classes to represent timestamps results in precision loss. To measure the precision loss, we compute observational errors, which is the average difference between actual timestamps and the converted class timestamps. Figure 15 shows the mean absolute error (MAE) and the observational errors for different partitions of classes. As a part of MAE, observational error are inherent such that it persists even with perfect class predictions (Conforti et al., 2020). Specifically, a small number of classes induce larger MAE, which is reasonable since converting actual timestamps to coarser time-span classes introduces larger additional observational errors.

However, this does not imply that extremely fine partitions should always be used to reduce observational error. We find that finer class partitioning, such as 144 classes, does not improve the performance. In particular, figure 16 presents the performance of the TICL model on the TOC test set under different class partitioning. The overall distribution of predictions exhibits similar patterns despite different granularities. Figure 17 highlights both class accuracy and hour accuracy for the model. The visualisation shows that while class accuracy drops significantly as the number of classes increases, the overall hour accuracy remains stable once the number of classes exceeds 24. This degradation in class accuracy with finer partitioning can be attributed to the smaller sample volumes within each class, as illustrated in figure 18. The smaller the sample volume for each class, the more underrepresented it tends to be (Sangalli et al., 2021). This suggests a potential drawback of finer class partitioning.

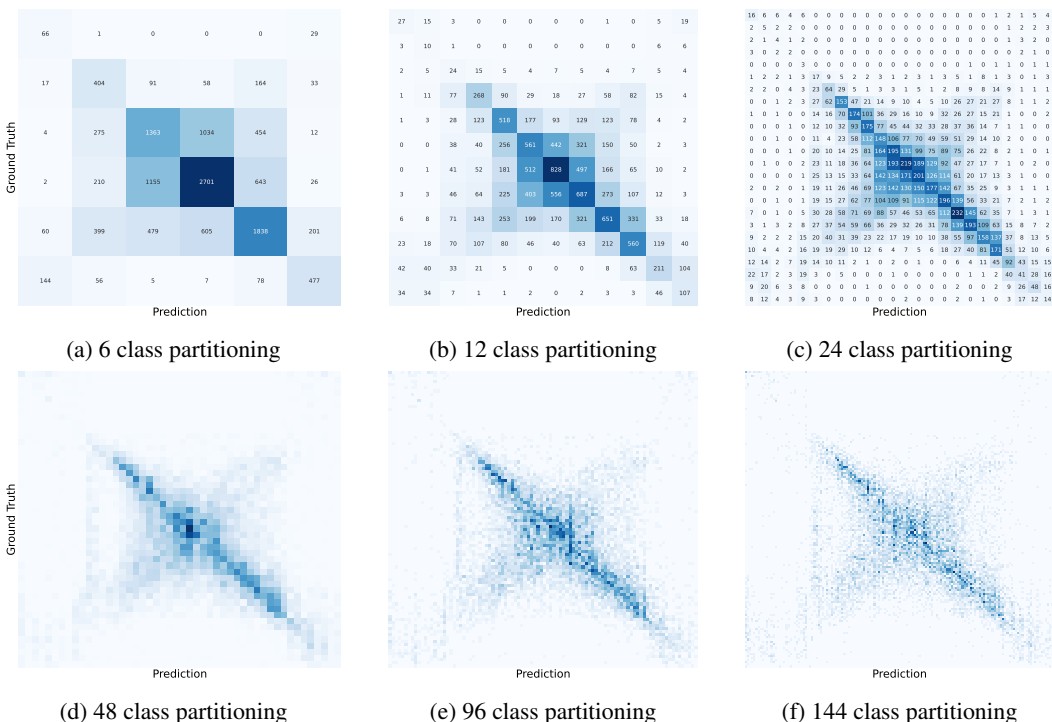

(a) 6 class partitioning    (b) 12 class partitioning    (c) 24 class partitioning

(d) 48 class partitioning    (e) 96 class partitioning    (f) 144 class partitioning

Figure 16: **Confusion matrices under different number of classes** provide more in-depth comparison of timestamp estimation performance.

Since the difference between timestamp estimation performance of the 24-class partition and the optimal result achieved with different class partitioning is within an acceptable range, we choose the 24-class partition as the default in our main work. This choice allows for a fair comparison against previous methods, to ensure that our improvements are due to the proposed techniques rather than variations in class partitioning. Additionally, the 24-class partitioning, which reached Class Accuracy $\approx$ Hour Accuracy, also ensures that each class can be assigned enough samples so that a robust time representation could be learned.

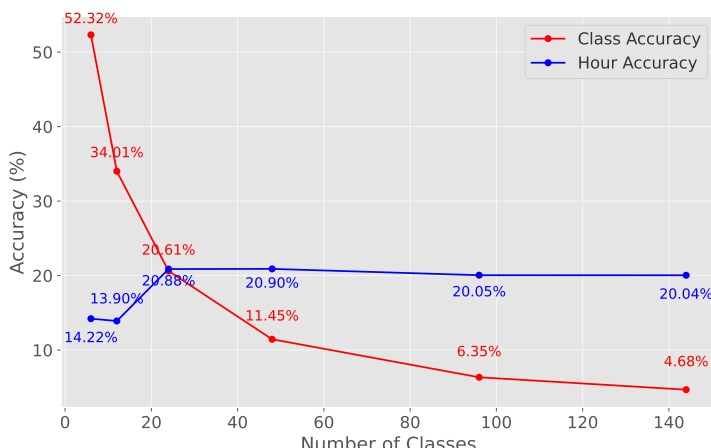

Figure 17: **Accuracy under different class partitioning**, Class Accuracy represents the raw classification accuracy, and Hour Accuracy is calculated by $\frac{1}{|D|} \sum_{i=0}^{|D|-1} \mathbf{1}_{\left( \left\| \hat{Y}_i - Y_i \right\|_1 \leq 30 \text{ minutes} \right)}$, $\hat{Y}_i, Y_i$ are prediction and ground truth timestamps correspondingly.

To sum up, the ablation study on number of classes indicates that while the proposed TICL method can easily be extended to finer class partitioning schemes and maintains good hour accuracy and MAE, moderate granularity in class partitioning yields the best results for timestamp estimation tasks. This supports our choice of a 24-class partitioning scheme for consistent benchmarking to previous baselines and verification of our conjecture on robust time-aware representation learning.

## A.4 Implementation details of the proposed TICL

In section 3.2, we covered the high-level design of the TICL model we devised to learn time-aware representations. This section provides additional details for the proposed TICL model.

### A.4.1 Model details

**Time Encoder:** The Time Encoder consists of several fully-connected layers, with the detailed architecture shown in figure 19. The objective of the Time Encoder is to transform the raw timestamps into meaningful time-aware representations that are aligned with image representations. To achieve the goal, the raw timestamps are first preprocessed into 24 one-hot class embeddings. The Time Encoder then takes these input class embeddings and projects them to the desired representation space.

**Image-Time Adaptor module:** The Image-Time Adaptor module is employed to adapt the raw backbone features with Time Encoder outputs, as depicted in figure 19. Training the Image-Time Adaptor module and Time Encoder jointly using a contrastive learning scheme allows for effective alignment between the two modalities.

**Hyper-Parameters:** We use an Adam optimiser with an initial learning rate of $5 \times 10^{-4}$ and a weight decay of $1 \times 10^{-6}$. The training process spans 20 epochs, with the learning rate halved every 2 epochs and a batch size of 512. The temperature parameter is initialized to 0.07. All input images are resized to $224 \times 224$.

### A.4.2 Computational efficiency

Since the majority part of the TICL model, the CLIP image encoder, is frozen during training, the TICL training is thus efficient with a small number of trainable parameters. Figure 20 shows that TICL achieved the best performance with the minimum trainable parameters among existing

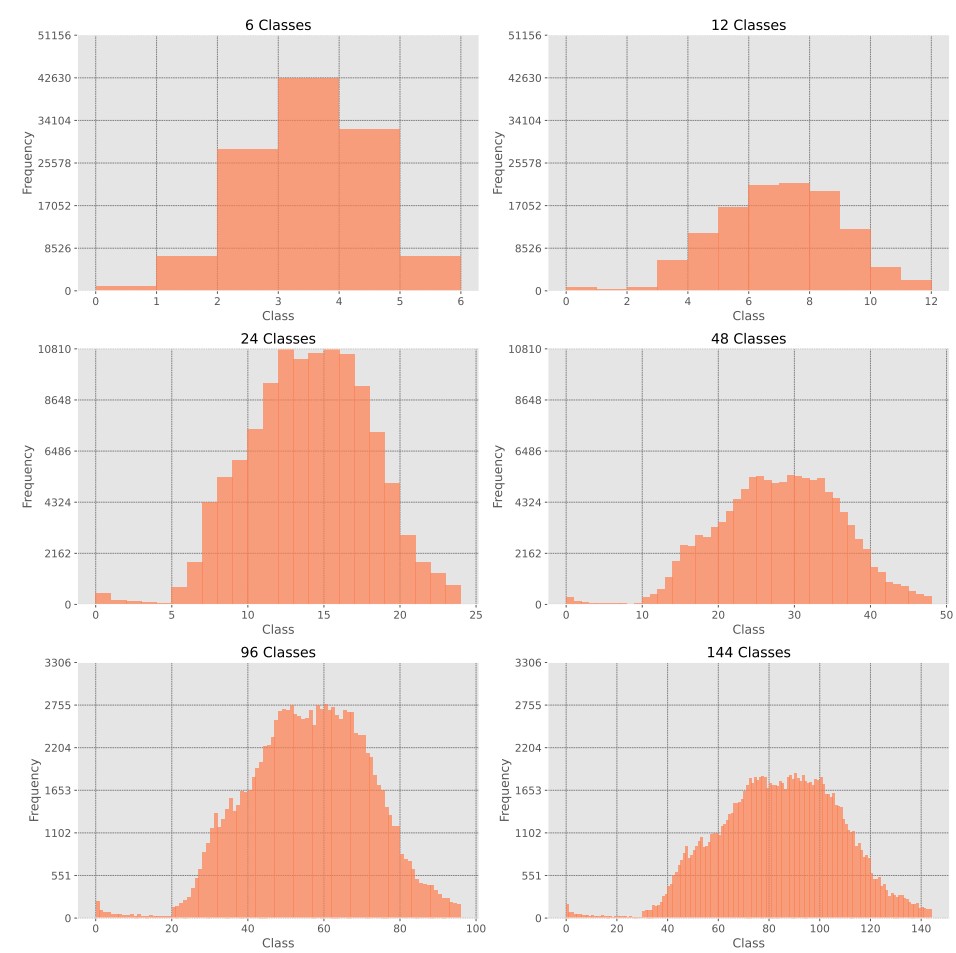

Figure 18: **Class sample distribution under different class partitioning schemes.** As the number of classes increases, classes are assigned with fewer samples.

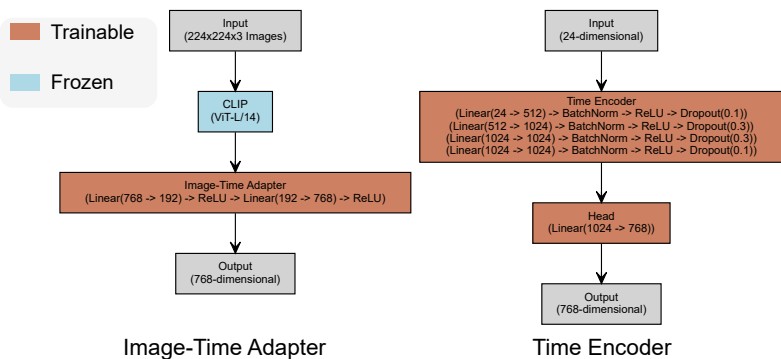

Figure 19: Visualisation of architectures of TICL sub-modules.

methods. Benefiting from the fewer trainable parameters, training on precomputed image features is significantly faster. Also, in figure 21, we show that simply scaling up the model parameters for previous works may even degrade the performance. We suspect that it is due to the more severe overfitting of the larger models on training samples. In comparison, TICL model reached better performance with moderate total number of parameters.

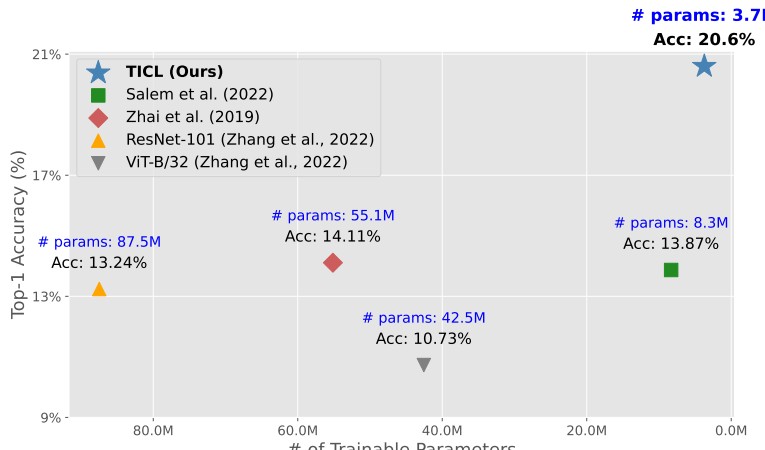

Figure 20: **Trainable parameters and performance comparisons.** It shows that our method outperforms the previous methods with minimal trainable parameters.

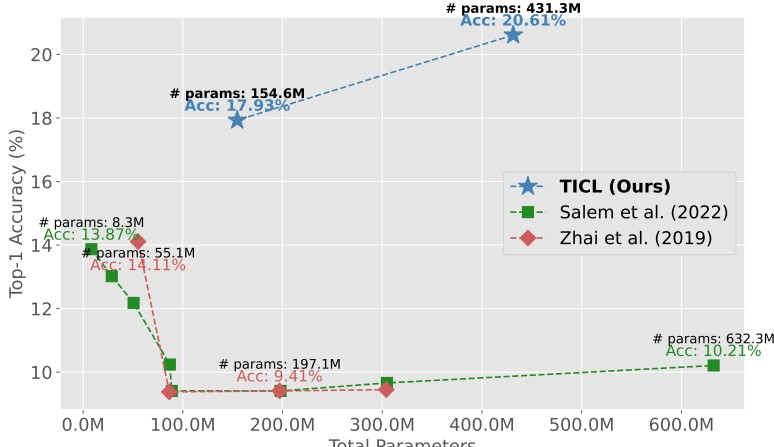

Figure 21: **Total parameters and performance comparisons.** It revealed that the TICL model maintains better performance with a moderate total number of parameters.

### A.4.3 TIMESTAMP ESTIMATION INFERENCE PIPELINES

Two different inference pipelines were devised. The first pipeline, shown in figure 22b, adheres to the classification scheme, selecting the timestamp with the highest similarity within a finite timestamp representation pool encoded from $C$ one-hot embeddings. The second pipeline, shown in figure 22c, converts the problem to a retrieval-style formulation, using known image-timestamp pairs from the training set. The model returns the class-level timestamp of the most similar samples to it in the training set using an efficient vector search engine (Johnson et al., 2019).

### A.5 DOWNSTREAM TASK PIPELINES

Before diving deeper into the analysis of each downstream task, we would like to delineate implementation details of the downstream task pipelines. Figure 23 depicts the downstream task pipelines for time-based image retrieval in section 5.3.1 and video scene classification in section 5.3.2. Figure 24 illustrates details for time-aware image editing task. Specifically, We freeze all parameters of the TICL model throughout the downstream tasks, enabling a direct evaluation of the potential of time-aware representations in a zero-shot manner.

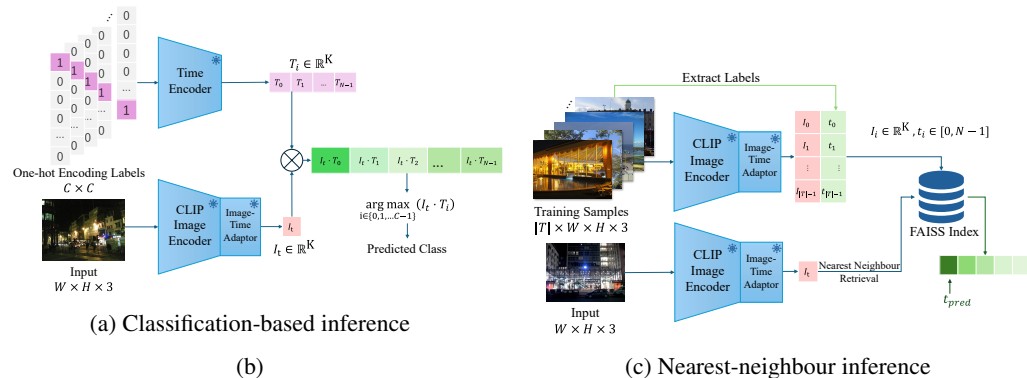

(a) Classification-based inference

(b)

(c) Nearest-neighbour inference

Figure 22: **Detailed illustration of different inference pipelines.** In (a), the model selects the timestamp with the highest similarity to the input image from a finite set of pre-encoded one-hot timestamp representations. (b) shows that the model estimates timestamp by finding the corresponding timestamp of the nearest-neighbour to the input images from the training set based on the TICL representations.

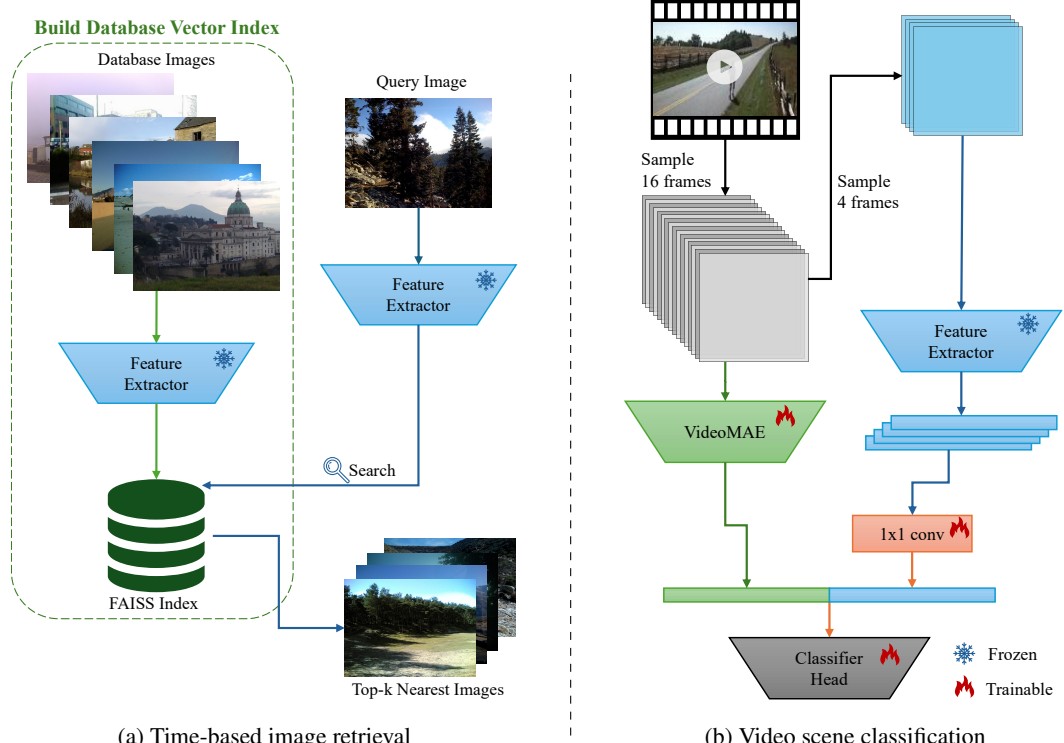

(a) Time-based image retrieval

(b) Video scene classification

Figure 23: **Zero-shot downstream pipelines.** (a) corresponds with experiment pipelines for section 5.3.1 which is a zero-shot vector search engine for same-hour images based on FAISS (Johnson et al., 2019), and (b) shows the pipeline for section 5.3.2, in which we test the capabilities of time-aware representations by pluging in the corresponding models to the feature extractor whose outputs are convoluted and concatenated to the backbone features (Tong et al., 2022).

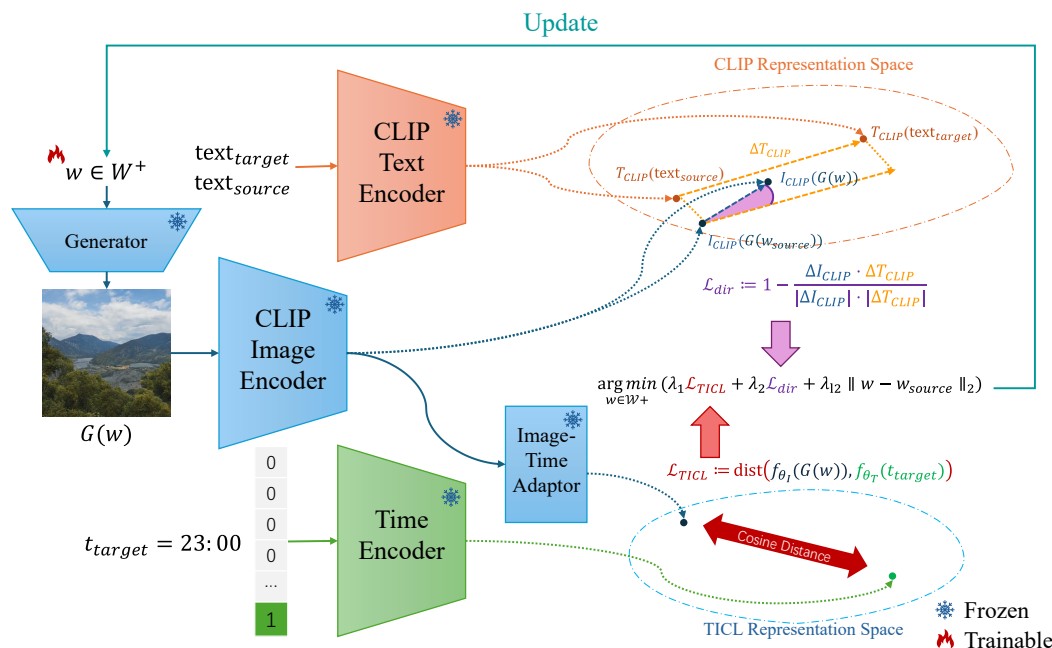

Figure 24: **Time-aware image editing pipeline.** This is the pipeline for section 5.3.3, where $w, w_{source}$ represents latent vectors for ongoing edit outcomes and original images, $t_{\text{target}}$ is the one-hot encoding of the desired time of day for the output image, $G(\cdot)$ is the generator, $\text{dist}(\cdot, \cdot)$ computes the cosine distance between two input representations, $\Delta I_{\text{CLIP}}$ is the difference between CLIP embeddings of the original image, $\Delta T_{\text{CLIP}}$ stands for the difference between the source and target caption embeddings. $f_{\theta_I}(\cdot), f_{\theta_T}(\cdot)$ corresponds to components of the TICL model.

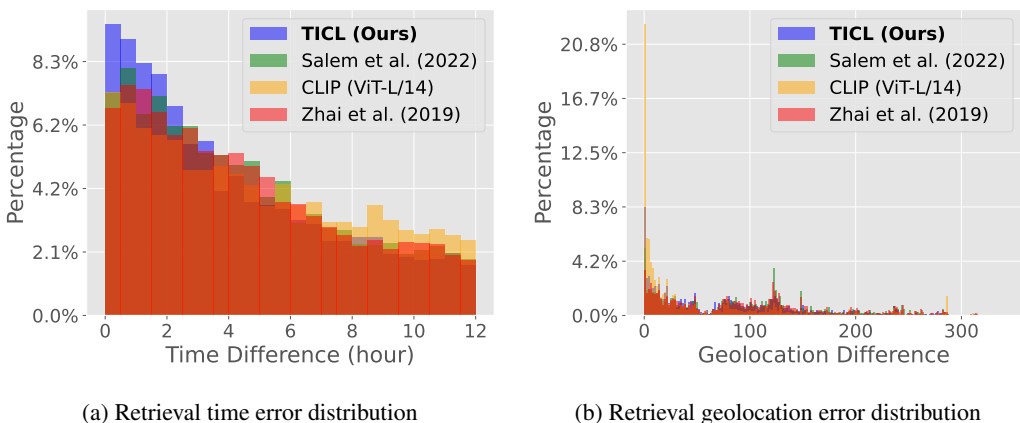

(a) Retrieval time error distribution

(b) Retrieval geolocation error distribution

Figure 25: **Comparison of Geo-temporal error distribution.** It is collected among top-100 retrieved images using different representations.

### A.6 TIME-BASED RETRIEVAL

#### A.6.1 QUALITATIVE RETRIEVAL RESULTS

Figure 26 provides a closer look at the retrieved images using the pipeline in figure 23a as part of a more detailed qualitative evaluation of retrieval performance. Some of the retrieved images have totally different content from the query images, but share similar light conditions. This suggests that our model disentangles the time-aware representations from CLIP representations, which have more

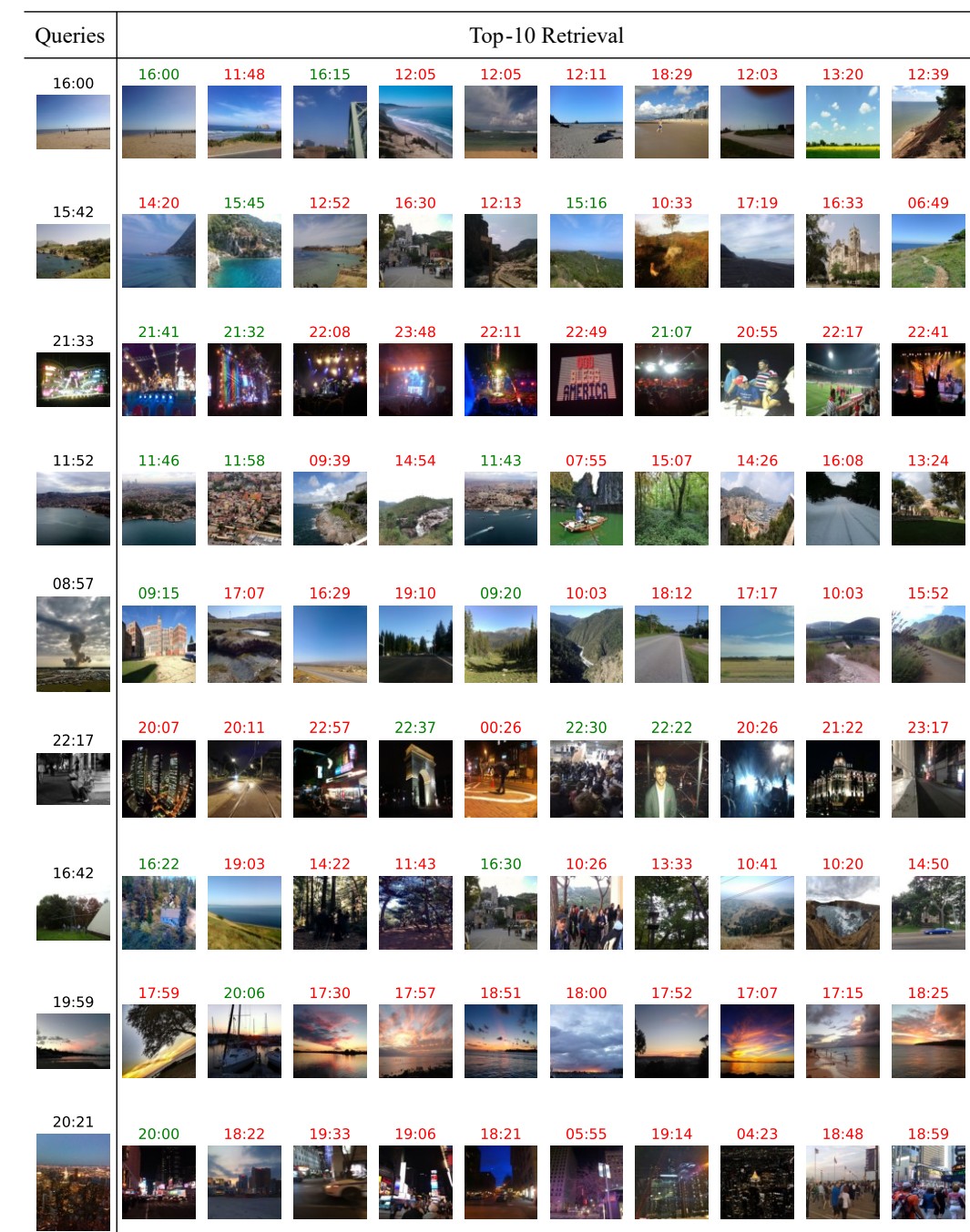

Figure 26: **Randomly sampled retrieval results.** Each image is annotated with its corresponding timestamp, green captioned images are positive retrieval while red are negative predictions with Error > 00:30, retrieved images closer to the left have larger similarity to the query images.

semantic focus to the subjects. In addition, the negative predictions still share similar illumination to the query images, highlighting the visual essence and ambiguity of time-aware representations.

### A.6.2 RETRIEVAL ERROR DISTRIBUTION

We also analyzed the distribution of metadata differences between the retrieved images and their corresponding query images. Specifically, Figure 25a illustrates the distribution of timestamp errors

among the top-100 retrieved samples for different representations. The results show that TICL retrieves a higher percentage of images with smaller time errors compared to other representations, demonstrating its superior accuracy in time-based retrieval tasks.

Figure 25b further shows the geolocation error distribution. Images retrieved by vanilla CLIP representations are geographically closest to the queries, suggesting that CLIP represents a rich understanding of scene priors strongly related to geolocation. This capability was delineated in some previous Visual Place Recognition works using CLIP backbone (Keetha et al., 2024; Vivanco et al., 2023). We suspect that This contextual awareness is partly inherited by TICL representations, which achieved moderately better performance than other time-aware representations without using CLIP. From this observation, we suspect that TICL disentangled time-aware representations at the cost of compromising the understanding of other metadata attributes.

### A.7 VIDEO SCENE CLASSIFICATION

#### A.7.1 EXPERIMENT DETAILS

The performance of the model shown in figure 23b on the video scene classification task was evaluated across three datasets, each containing videos with distinct styles.

- **Hollywood2-Scene** (Marszałek et al., 2009) is a movie clip-based dataset with 570 training videos and 582 test videos across 10 scene classes, totalling 20.1 hours. Each video represents a specific dramatic scene with multiple shots, meaning drastic view/subject changes within.
- **YUP++** (Derpanis et al., 2012) comprises 1200 videos across 20 scenes captured by either stationary or moving cameras. Given the significant differences between the 20 scenes and the fact that the average clip duration is only 5 seconds, the classification task on it is considered less challenging (Wang & Koniusz, 2023).
- **360+x** dataset (Chen et al., 2024) is a more recent dataset introduced for holistic dynamic scene understanding with multiple views captured by stationary cameras. It consists of 15 indoor scenes and 13 outdoor scenes, with 1380 clips totalling 67.78 hours. Its multi-view and stationary camera traits enable us to evaluate how our pretrained time representations perform on different types of views individually.

For fair comparison, a fixed set of hyperparameters was used in different experiment trials. Apart from the number of epochs and the learning rate, we followed all the parameter settings in Tong et al. (2022). We report the best result achieved for each method tested. Specifically, a training/validation split of 5:1 was applied to each original training dataset to fairly select the best checkpoints for each method.

#### A.7.2 TIME-AWARE REPRESENTATIONS ON VIDEO FRAMES

As discussed in section 5.3.2, the observed improvements when integrating time-aware features with video classification backbone models could be attributed to the stronger intra-video consistency of these time-aware features.

To provide quantitative evidence of this consistency, we examine the characteristics of time-aware features across frames within each video. The backbone VideoMAE (ViT-B) model takes the input by sampling 16 frames evenly from each video. For the 16 input frames, we observed that, the time-aware features of these 16 frames exhibit significantly smaller average variance compared to their CLIP features, as shown in table 4.

This finding supports our intuition that a natural video that depicts a dynamic scene is typically captured over a short period of the day, leading to relatively small changes in the time-aware features of consecutive frames. In contrast, the CLIP features show more drastic changes between frames, making it is harder to summarise consistent frame-wise features into coherent video-level features. The t-SNE visualisation comparisons to these features in figure 5 and figure 27 reveal that TICL representations are more separable than raw CLIP features on various datasets.

Thus, time-aware feature extractors provide more consistent representations across different frames, making it easier to capture time-related visual priors in videos, which correlate with scene categories.

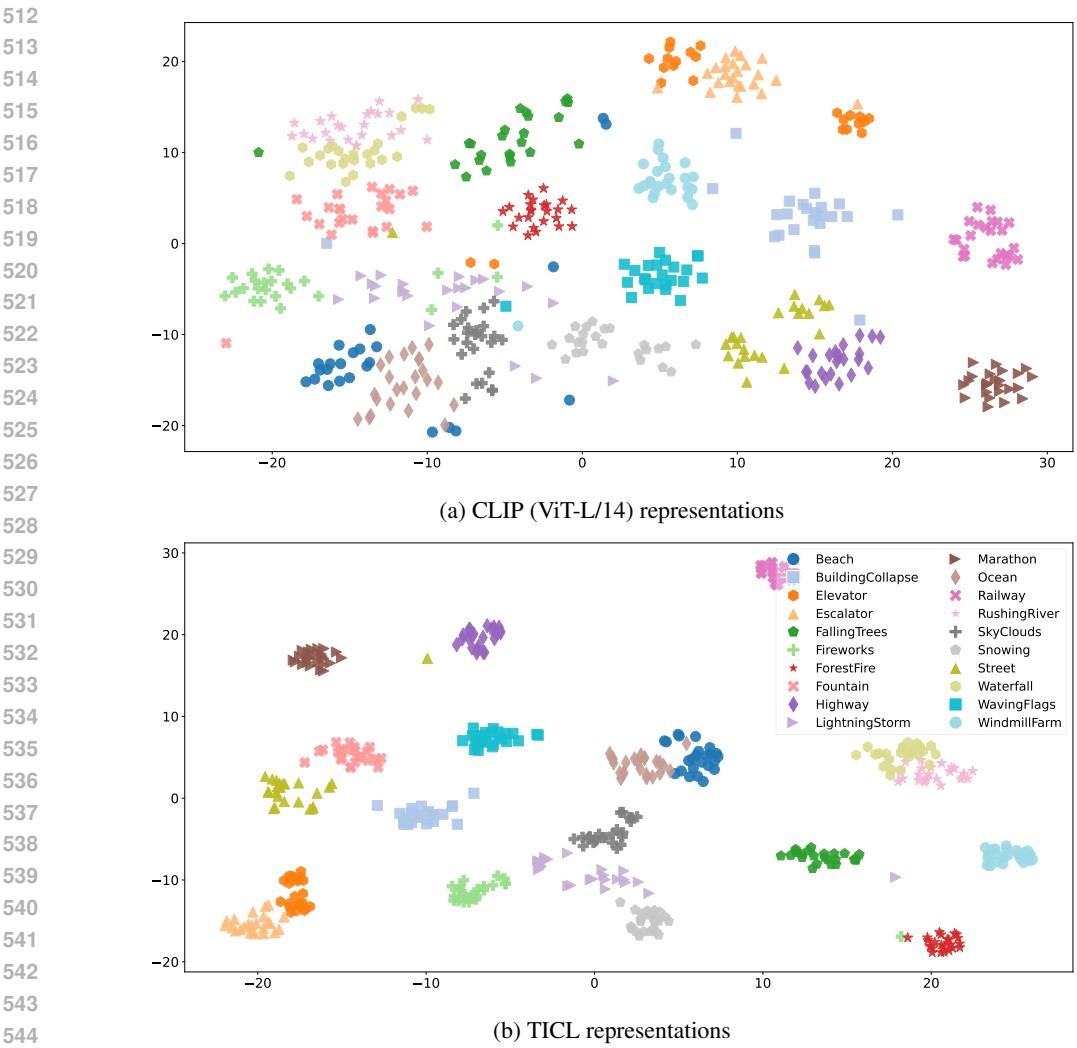

(a) CLIP (ViT-L/14) representations

(b) TICL representations

Figure 27: **t-SNE visualisation comparison.** It visualises time-aware video representations in YUP++ dataset (Derpanis et al., 2012). Each representation is annotated by their corresponding labels. It exhibits a similar trend to figure 5.

Table 4: **Mean intra-video feature variance.** It is computed by the mean feature variance of 16 input frames for each video using different models, showing a quantitative evidence of intra-video feature consistency of time-aware representations.

| Representation | Hollywood2-Scene | YUP++ | 360+x (Third-person) | 360+x (Panoramic) |
|---|---|---|---|---|
| CLIP (ViT-L/14) | $7.49 \times 10^{-2}$ | $2.49 \times 10^{-2}$ | $3.31 \times 10^{-2}$ | $2.83 \times 10^{-2}$ |
| Salem et al. (2022) | $3.52 \times 10^{-6}$ | $1.23 \times 10^{-6}$ | $7.55 \times 10^{-7}$ | $7.86 \times 10^{-7}$ |
| Zhai et al. (2019) | $2.50 \times 10^{-4}$ | $1.00 \times 10^{-4}$ | $8.50 \times 10^{-5}$ | $7.59 \times 10^{-5}$ |
| TICL (Ours) | $3.33 \times 10^{-4}$ | $1.24 \times 10^{-4}$ | $1.44 \times 10^{-4}$ | $1.33 \times 10^{-4}$ |

These time-aware video priors eventually improved the video scene recognition performance as illustrated in table 3.

However, it is observed that the representations in Salem et al. (2022) and Zhai et al. (2019) have much smaller intra-video feature variances, but they perform worse than the TICL features we proposed. Given that the previous methods produce 128-dimensional time-aware representations, which dimensionality is much lower than TICL representation, it is expected that they have much smaller variances. Moreover, although previous methods perform moderately better than the baseline meth-

Table 5: **FID Scores.** They quantitatively show how realistic the image editing results are for different methods on two image edit directions.

| Methods | Day-to-Night ↓ | Day-to-Sunset ↓ |
|---|---|---|
| Latent optimisation ($\mathcal{L}_{CLIP}$) (Patashnik et al., 2021) | 53.55 | 50.60 |
| Latent optimisation ($\mathcal{L}_{dir}$) | 50.07 | 50.59 |
| **Latent optimisation ($\mathcal{L}_{dir} + \mathcal{L}_{TICL}$)** | **48.97** | **50.41** |
| StyleGAN NADA (Gal et al., 2021) | 78.80 | 66.58 |
| CLIPStyler (Kwon & Ye, 2022) | 71.12 | 73.59 |

Table 6: **User study evaluating image editing qualities,** in which we report preference scores and their standard deviation (in brackets). Preference scores range from 1-5, and higher scores mean better preferences.

| Methods | Day-to-Night ↑ | Day-to-Sunset ↑ |
|---|---|---|
| Latent optimisation ($\mathcal{L}_{CLIP}$) (Patashnik et al., 2021) | 2.80 (0.60) | 2.84 (0.53) |
| Latent optimisation ($\mathcal{L}_{dir}$) | 2.63 (0.85) | 3.28 (0.67) |
| **Latent optimisation ($\mathcal{L}_{dir} + \mathcal{L}_{TICL}$) (Ours)** | **3.34 (0.64)** | **4.01 (0.58)** |
| StyleGAN NADA (Gal et al., 2021) | 2.41 (0.89) | 2.36 (1.17) |
| CLIPStyler (Kwon & Ye, 2022) | 2.08 (0.62) | 1.81 (0.93) |

ods in the majority of test datasets, their performance degradation in panoramic video datasets suggests a limitation in terms of generalisation ability between different styles of videos, especially for those captured in rare camera views in the 360+x dataset (Chen et al., 2024). In contrast, TICL utilising a strong foundation model generalised better across different kinds of videos.

In summary, time-aware representations could provide a more coherent representation among multiple sequential frames in a video, which are relatively invariant to sudden view/object changes altering the semantic meaning of the frame. Among the time-aware representations, TICL representations give more robust time-aware priors that generally bring more improvements than all the other time-aware representations on different styles of video.

## A.8 TIME-AWARE IMAGE EDITING

### A.8.1 LATENT OPTIMISATION

Figure 24 gives an overview of the experiment pipeline we used for the time-aware image editing task. Additional results of latent optimisation based editing are presented. We varied the initial latent vectors and target hours to show the broader capabilities of our approach. Figure 28 provides more examples of time-aware image editing with intermediate results during optimisation steps. The results suggest that our method could be applied to broad time-aware editing directions, which can start from images from various times of day.

In addition to the qualitative evaluation results, we also include quantitative metrics to evaluate the synthesis results. Table 5 gives FID scores (Heusel et al., 2017) to different edit directions calculated by the official PyTorch implementation by Seitzer (2020). Our method outperforms existing methods with a smaller FID score suggesting more realism in the synthesised images. Additionally, we conducted a user study (by using the mean-opinion-score scheme) on the output images. The preference scores for each method are reported in table 6, further demonstrating the advantages brought by incorporating time-aware representations, which led to more favourable results.

### A.8.2 EDITING DIFFUSION MODELS

Given that the previous baseline latent optimisation image editing method has limited capabilities, we extended our experiment to a more recent editing method Parmar et al. (2024) using diffusion models (Ho et al., 2020; Rombach et al., 2021). Specifically, we optimise the edit target text embedding $E_{text}^*$ to minimise the cosine distance between the time-aware representation of the output images and the target timestamp representations, which is written as:

$$E_{text}^* = \arg\min_{E_{text}} \text{dist}\left(f_{\theta_I}\left(G\left(x, E_{text}\right)\right), f_{\theta_T}\left(t_{target}\right)\right)$$

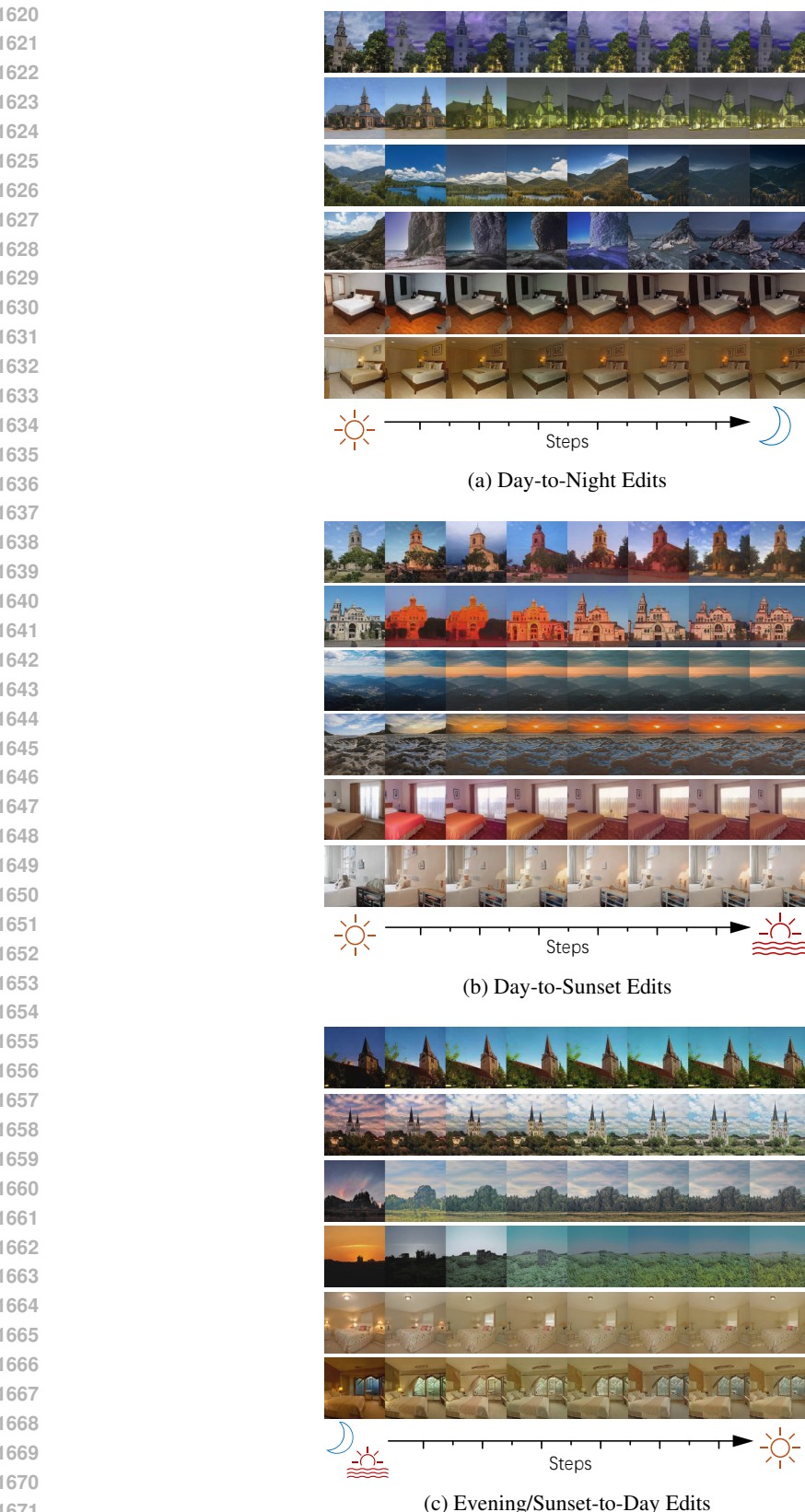

Figure 28: **Time-aware image editing examples through latent optimisation.** This figure shows the progression of edits from various starting points to target times of day.

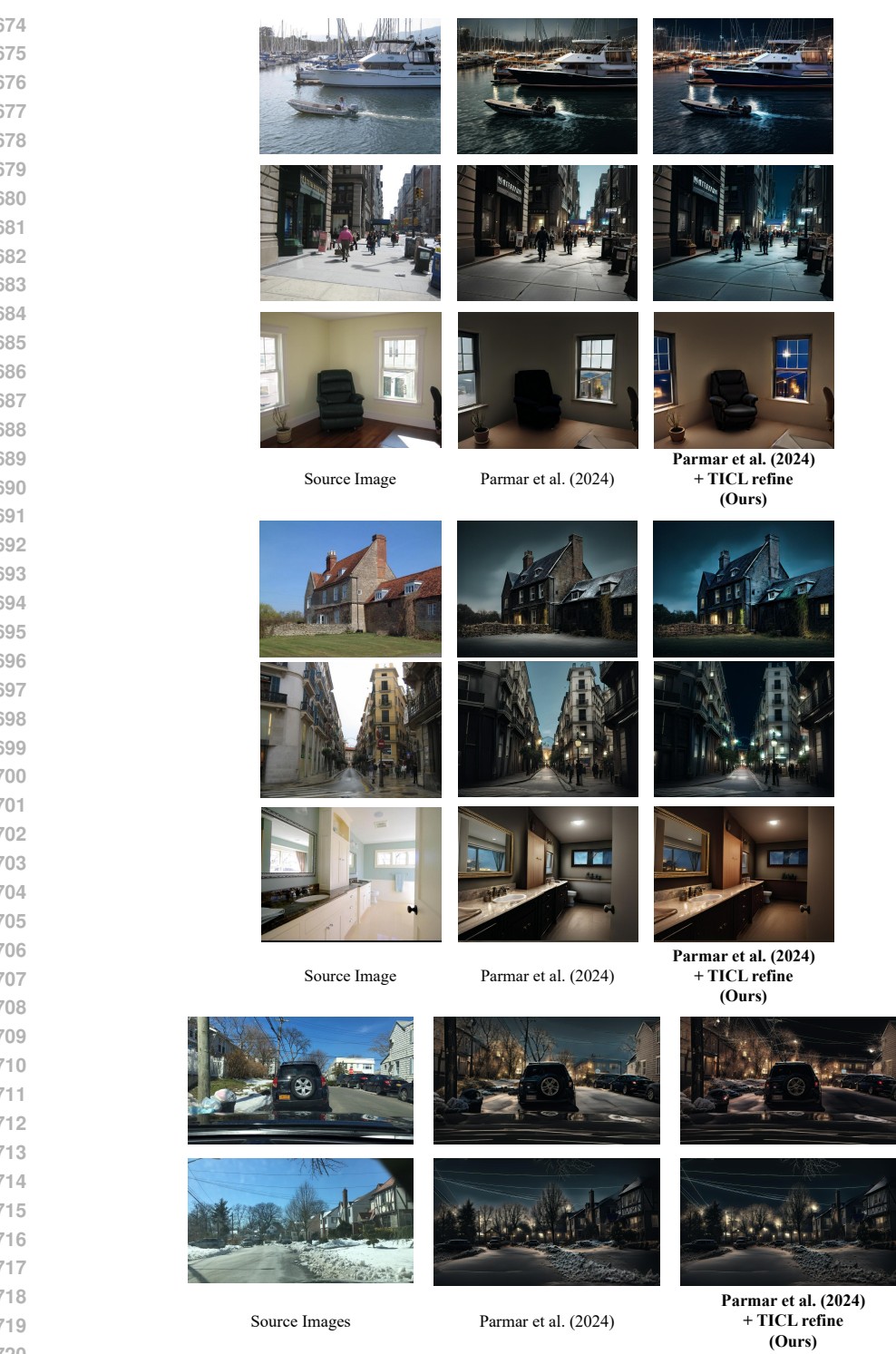

Figure 29: **Visualisation of Day-to-night edits.** This figure presents the results of time-aware editing using diffusion model (Rombach et al., 2021; Parmar et al., 2024), specifically transitioning images from day to night.

where $E_{text}^*$ is the target text representation for the text-based image editing model $G(\cdot, \cdot)$ takes input image $x$ and guidance text embedding $E_{text}$. $f_{\theta_I}, f_{\theta_T}$ corresponds to TICL model components.

$dist(\cdot, \cdot)$ measures the cosine distance of two representations. It essentially optimises the guidance text representation $E_{text}$ to achieve better editing results that visually align with the target time.

As shown in figure 29, although additional optimisation steps for each edit are required, it refines the existing method with more reasonable synthesis results compared with using purely text editing guidance, further proving the general applicability of the TICL representation to the whole image-editing subfield.

### A.9  Text queries on time-aware representations

In section 5.3.2, we explored the semantic correlations between timestamps and scenes, and here we provide several examples to illustrate these connections. The Time Encoder and Image-Time Adaptor modules are designed to align visual CLIP representations and time representations. As a result, the learned time representations naturally align with CLIP text representations. This alignment allows us to factorise text concepts using TICL time representations and vice versa. To do this, specifically, for each input text embedding, $T_{CLIP}$, we compute the similarity with time-class embeddings, $T_i$, using the softmax function:

$$\textbf{Softmax} = \frac{\exp\left(T_{CLIP} \cdot T_i\right)}{\sum_{j=0}^{|C|-1} \exp\left(T_{CLIP} \cdot T_j\right)}$$

where $T_i, T_j$ are the TICL class embeddings. This formulation offers a probabilistic measure of the similarity between text representations and time classes. The resulting 24-hour class probabilities are shown in figure 30.

The results clearly demonstrate that texts describing specific times of day are directly associated with corresponding time periods. Beyond this direct relationship, we also observe indirect associations. For example, the word "breakfast" is by definition related to morning hours, while "thief" is often associated with nighttime activities. These uneven probability distributions across the 24-hour time-line reflect the natural relations between certain events, scenes, or concepts and their corresponding time periods.

However, some irregular trends in the probability distributions indicate that our time-aware representations, learned from a limited image dataset, still have room for further improvements, particularly for night-time concepts and images, where samples are rarer. This highlights the need for further refinement of the model to achieve more robust performance across all time periods.

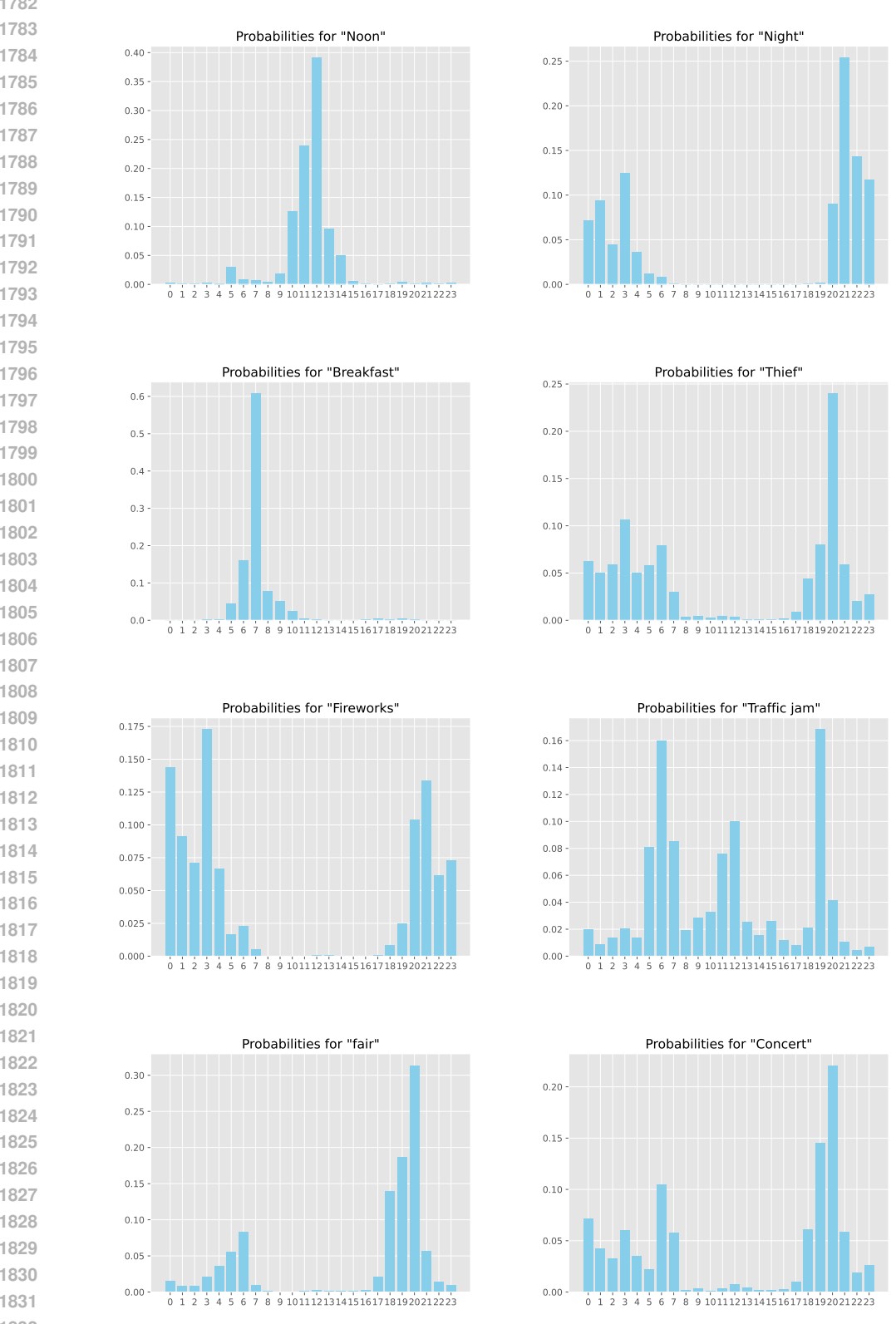

Figure 30: **Probability measure of the similarity between time classes and text queries.** The x-axis is hour classes and y-axis is probabilities calculated by **Softmax**.

