# OpenReview forum: "What Time Tells Us? Time-Aware Representation Learning from Static Images"
_ICLR.cc/2025/Conference — ICLR 2025 Conference Withdrawn Submission_

### Official Review · Reviewer_yVcu · 2024-10-25

**Soundness:** 2
**Presentation:** 3
**Contribution:** 1
**Rating:** 3
**Confidence:** 4

**Summary:**

This paper demonstrates that learning time-of-day classifier on top of a frozen CLIP backbone does better than learning it on top of other feature representations (e.g. DINO) as well as better than previous works that learned such models end-to-end (e.g. with a ResNet). To present a more faithful evaluation, the authors combine exiting datasets and manually filtering them to remove unnatural samples and samples with incorrect timestamps. In addition, the authors demonstrate that the resulting projection of the CLIP features can be useful for some other tasks that can benefit from time-of-day understanding (e.g. video scene classification).

**Strengths:**

The paper is relatively well written and easy to parse.

The conclusion that CLIP features are useful for time-of-day classification due to their strong semantic understanding capabilities is reasonable.

The proposed approach outperforms prior work by a large margin by capitalizing on CLIP features. That said, there are questions to the experimental setup (see below).

Cleaning up the annotations in existing time-of-day classification datasets is a useful effort.

**Weaknesses:**

Virtually no implementation and experimental setup details are reported in the paper, making it impossible to judge the significance of the reported results. Most importantly, it’s unclear if other methods were also trained on the clean training set collected by the authors or if the authors just evaluated the publicly available checkpoints. It is also unclear what the backbones used in ablation in Table 2 were pre-trained on (except for DINO-v2). It is also unclear why for some models ViT-Base variant is used, but for others (e.g. the CLIP backbone) ViT-Large is reported.

Same goes for the downstream task evaluations in section 5.3. For example, the proposed CLIP projection results in a major performance improvement on the Hollywood2-Scene dataset (26.8 accuracy points over the second-best variant) which is not explained by the authors and is probably an artifact of the (unreported) hyper-parameters used when learning a linear classifier on this dataset.

Overall, all the downstream evaluations in the paper are designed by the authors and the details are not reported so it’s impossible to trust the results.

The contribution is significantly overclaimed. The authors talk about "representation learning" but training a projection module on top of a frozen CLIP encoder is not representation learning. The only (somewhat) convincing results are reported on the task of time-of-day classification for which the projection module was trained.

To sum up, the focus of this paper is extremely narrow, the novelty is minimal, and the experimental evaluation is flawed/unconvincing.

**Questions:**

Please report:

The exact training dataset used for each compared method.
Pre-training details for all backbones used in ablations.
Rationale for using different model sizes (ViT-Base vs ViT-Large) across ablation experiments.
A detailed description of each downstream task evaluation setup.
Potential reasons for such a large performance improvement gap between Hollywood2-Scene and other video scene classification datasets. Conduct additional experiments or analysis to verify that the improvement is not due to some artifact of the setup.

Please revise the claims to more accurately reflect the scope of the work.

---

> ### Author Response · Authors · 2024-11-14
> **Author Response**
>
> Dear Reviewer yVcu,
>
> Thank you for your detailed and constructive feedback, which is very helpful in improving our manuscript. However, we have to respectfully disagree with part of your statement. Here is our responce to the review:
>
> ### 1. Experiment Details
>
> You noted that "virtually no implementation and experimental setup details are reported." We respectfully disagree with this accusation. These details are either included in the provided code and are also discussed in Sections **A.4**, **A.7.1**, and **A.8** of the supplementary material. We will revise the manuscript to more explicitly reference these sections to ensure clarity.
>
> Besides, we confirm that all results in **Table 1** are from models trained on our cleaned dataset (the TOC dataset) as we stated in line 279, ensuring a fair comparison. This point will be explicitly clarified in the revised manuscript.
>
> To further improve transparency, we will expand the manuscript with additional experimental details, including hyperparameter configurations and their influence on results. Additionally, we will gradually release more codes and experimental setups. We are happy to provide any additional details you may find unclear.
>
> ### 2. Model Choices for Video Scene Classification
>
> We used **ViT-Base/16** variant of VideoMAE because it is the default backbone for VideoMAE, while **CLIP ViT-Large/14** was chosen for other tasks due to it's also default choice in the previous work.
>
> - These models are not compared against each other; they are combined for video scene classification tasks, leveraging their respective strengths without artificially aligning configurations.
>
> ### 3. Pretraining Details
>
> All backbones are pretrained (on ImageNet or any other default pretraining dataset) as indicated in the footnotes of **Table 2** on line 340. unless otherwise specified. For clarity, we will explicitly state this in the revised manuscript to prevent confusion.
>
> ### 4. Performance Gap in Hollywood2-Scene
>
> Thank you for notifying us about the gap in Hollywood2-Scene performance results of VideoMAE + CLIP we reached at the original learning rate. After some further trials with lower learning rates, we obtained new reasonable results. Below are the summarized training configurations:
>
> | Dataset    | Learning Rate | Epochs | Batch Size | Acc (VideoMAE + Salem et al. 2022) | Acc (VideoMAE + Zhai et al. (2019)) | Acc (VideoMAE + CLIP) | Acc (VideoMAE + TICL) |
> | ---------- | ------------- | ------ | ---------- | ---------------------------------- | ----------------------------------- | --------------------- | --------------------- |
> | Hollywood2 | 1e-4 → 5e-5   | 20     | 2          | 32.99% → 45.53%                    | 32.65% → 51.03%                     | 22.51% → 52.92%       | 59.79%→56.53%         |
>
> We will also update all the related results to the change in this hyperparameter.
>
> We appreciate your thoughtful comments and will revise our paper to improve clarity, accessibility, and transparency.

---

### Official Review · Reviewer_jWGw · 2024-10-29

**Soundness:** 3
**Presentation:** 3
**Contribution:** 2
**Rating:** 5
**Confidence:** 5

**Summary:**

This paper presents a dataset (TOC: Time-Oriented Collection) and method (TICL: Time-Image Contrastive Learning) for hour prediction. The TOC dataset is a filtered subset of the Flickr images from the CVT dataset. Filtering is done by removing irrelevant images, like memes/text and images with incorrect timestamps. TICL is a method that aligns CLIP image embeddings after an adapter layer with the embeddings from a time encoder using a CLIP-like contrastive loss. The experimental results show state-of-the-art performance on hour prediction compared to other methods, as well as applications of the time-aware image embeddings on retrieval, editing, and video scene classification.

**Strengths:**

TICL achieves SoTA performance on hour prediction compared to other methods, such as Zhai et al. (2019) and Salem et al. (2022).

The authors conduct ablations using different image backbones, clearly showing that CLIP is the best option. Furthermore, the authors show that using the time encoder module and time adapter performs better on most metrics.

The paper has an interesting analysis explaining why regression methods don’t work well, even when trained with a circular MSE loss.

Cleaning the images from CVT makes sense, given that some of them do not contain any time information (for example, memes), and training with them would probably hurt the model performance.

The time-based image retrieval application with TICL shows significant improvement over other methods.

**Weaknesses:**

Method and results

TICL is only able to predict the hour of the day, while previous SoTA methods are able to predict multiple things besides hours. The method proposed by [1] predicts the hour, month, and geographical location of the images, while [2] predicts the hour, week and month. Comparing TICL with these other methods is not completely fair, given that they need to allocate capacity to other tasks as well.

Other recent methods, such as [3] are also able to predict hours and months indirectly and are trained with similar datasets, but the authors didn’t include it in their evaluation. The code and model weights for [3] are publicly available and the authors should include it as a baseline. Please include this model as a baseline.

The method itself is not very novel. It is largely based on GeoCLIP with simplified components, such as replacing the Random Fourier Features (RFF) Encoder with an MLP and removing the dynamic queue.

The authors should conduct ablations with different time representations and encoder architectures. There should be a table comparing their time encoder with the RFFs encoder from GeoCLIP and with Time2Vec [4].

It’s not clear why the hours are converted into one-hot encoded vectors before passing them to the time encoder. A more straightforward approach would be to pass the hour directly as an integer/float and project it to a high-dimensional vector with a linear layer. Another option would be to decompose the hour into sine and cosine components, similar to [5]. Please conduct further ablations using this time representations.

It would be interesting to see a more in-depth analysis of the time prediction errors. The confusion matrices are a good start, but quantitatively, what is the accuracy at different moments of the day? In other words, how does the error during the morning, noon, afternoon, and night compare against each other? For example, it seems like in the AMOS test set a lot of images in the morning are being confused by images in the afternoon.

Also, one hour can look very different in the same location but different months, or in the same month but different locations. How does the time prediction error close to the Equator compare against a location at high latitudes? Or how does the time error in a location close to the tropics change during the summer and winter seasons. These questions are interesting but left unexplored.

Dataset

The AMOS subset from CVT has ~100k images. Since this dataset is from outdoor cameras across the whole day and year, around half of them are captured at night. In some cameras, these images look too dark to get any meaningful time information. However, this leaves around 50k daytime images, most of which have good weather and there is no reason to exclude them from the test set. If the authors only train on TOC, why are they testing the model only on 3556 AMOS images?

Cleaning the Flickr subset of CVT makes sense, but the authors should’ve conducted experiments training the model with the original “noisy” dataset and the clean dataset to show how this step is crucial for good time prediction.

Applications

The retrieval and editing applications are interesting, but it’s not clear why a time-aware time embedding would help in the video scene classification task. First of all, why would a time-aware embedding help in scene classification? Intuitively, a model for scene classification should be invariant to time, so why is TICL helping?

By looking at figure 5, it seems that TICL embeddings form better clusters than the vanilla CLIP embeddings for the different scene classes. However, most of the scene classes are indoors (bedroom, car, hotel, kitchen, etc.). The images from CVT are mostly from outdoor scenes, so how can the model help predict indoor scenes if it has seen very few indoor images? During training Also, the gap between VideoMAE+CLIP and VideoMAE+TICL seems unreasonably large compared to the other datasets, where gains are modest, why is that the case?

The time editing tasks seems to work well, but it would be interesting to see if it produces realistic shadows or color hues given the time of day. For example, a simple test would be to take a picture of an object with known height, let’s say at 10 AM and 4 PM, and measure the shadow lengths. Then, pass the 10 AM image to the editing model and change the time to 4 PM to see if the angle and length of the shadow in the generated image matches the real image.
References:

[1] Zhai, Menghua, et al. "Learning geo-temporal image features." arXiv preprint arXiv:1909.07499 (2019).

[2] Salem, Tawfiq, Jisoo Hwang, and Rafael Padilha. "Timestamp Estimation From Outdoor Scenes." (2022).

[3] Padilha, Rafael, et al. "Content-aware detection of temporal metadata manipulation." IEEE Transactions on Information Forensics and Security 17 (2022): 1316-1327.

[4] Kazemi, Seyed Mehran, et al. "Time2vec: Learning a vector representation of time." arXiv preprint arXiv:1907.05321 (2019).

[5] Mac Aodha, Oisin, Elijah Cole, and Pietro Perona. "Presence-only geographical priors for fine-grained image classification." Proceedings of the IEEE/CVF International Conference on Computer Vision. 2019.

**Questions:**

Questions

Please refer to the weaknesses section. Here are some additional questions:

Are all previous methods shown in table 1 retrained with the TOC train set?

During the dataset filtering process, the authors remove images that appear during daytime but are captured at 12 AM. Do they do the same for other typical night hours, such as 11 PM, 1 AM, etc.? Also, there might be some edge cases where 12 AM has sunlight, like in locations with high latitudes. Did the authors consider such cases?

What is the accuracy of the DBSCAN method in removing unnatural or uncalibrated images? If accuracy is not a good metric, how are the authors validating that the filtering method is working correctly?

---

> ### Author Response · Authors · 2024-11-14
> **Response part 1**
>
> Dear Reviewer jWGw,
>
> Thank you for your detailed feedback and constructive comments on our work. Your insights have been invaluable in identifying areas for improvement. Below, we address your concerns, provide clarifications, and outline revisions to further strengthen our manuscript.
>
> ### 1. **Comparison with Prior Work (Weaknesses, Q1)**
>
> We agree that comparing TICL with prior methods addressing multiple tasks (e.g., hour, month, and location prediction) is not entirely fair due to differing objectives and capacities. However, in previous works, it is widely acknowledged in ablations that predicting other metadata (month, geolocation, week) will not degrades but in contrast improves the hour prediction performance of previous works. e.g.
>
> - *Table 1, Page 6* of **Salem, Tawfiq, Jisoo Hwang, and Rafael Padilha. "Timestamp Estimation From Outdoor Scenes." (2022).**
> - *DenseNet-121 results of TABLE I, Page 5* of **Padilha, Rafael, et al. "Content-aware detection of temporal metadata manipulation." IEEE Transactions on Information Forensics and Security 17 (2022): 1316-1327.**
>
> Thanks for you advice of adding (Padilha, et al. 2022) as an additional baseline, despite different problem formulation, there are a few reasons for us to decide not to include it as a baseline:
>
> - Given the difference in the problem formulation, build reasonable evaluation metrics could be challenging (e.g. providing different input timestamp could have the result varies significantly).
> - We would have to retrain a model on TOC train dataset to avoid potential train/test leakage.
> - It's very similar to the baseline (Salem, et al. 2022) we have tested, if it is trained under the same construction to ours without satelite images and geolocation as additional inputs.
>
> Also, regarding results in Table 1, we want to clarify that:
>
> - **Yes**, all methods shown were retrained using the TOC training set for consistency. As we explicitly stated in the footnote of the Table 1 (line 279). We will emphasize this point with clearer statements in the revised manuscript.
>
> ### 2. **Time Representation and Ablations (Weaknesses)**
>
> We appreciate your suggestion to explore alternative time representations. We have conducted additional ablations with **Random Fourier Features (RFF)** and **Time2Vec (T2V)**, as shown in the table below, the performance comparisons of these techniques justified our design.
>
> | Image Encoder       | $f_{\theta_t}$ | $f_{\theta_{ITA}}$ | TOC Test Set: Top-1 Acc (%) ↑ | TOC Test Set: Top-3 Acc (%) ↑ | TOC Test Set: Top-5 Acc (%) ↑ | TOC Test Set: Time MAE (min) ↓ | AMOS Test Set: Top-1 Acc (%) ↑ | AMOS Test Set: Top-3 Acc (%) ↑ | AMOS Test Set: Top-5 Acc (%) ↑ | AMOS Test Set: Time MAE (min) ↓ |
> | ------------------- | -------------- | ------------------ | ----------------------------- | ----------------------------- | ----------------------------- | ------------------------------ | ------------------------------ | ------------------------------ | ------------------------------ | ------------------------------- |
> | **CLIP (ViT-L/14)** | RFF            | ✓                  | 16.75                         | 65.14                         | 46.61                         | 206.50                         | 6.07                           | 15.78                          | 22.27                          | 290.70                          |
> |                     | T2V            | ✓                  | 17.70                         | 45.69                         | 66.11                         | 185.89                         | 7.37                           | 21.74                          | 35.10                          | 264.25                          |
> |                     | Ours           | ✓                  | 20.61                         | 49.01                         | 67.83                         | 171.65                         | 13.55                          | 38.50                          | 57.28                          | 187.87                          |
>
> Continued...

---

> > ### Author Response · Authors · 2024-11-14
> > **Response part 2**
> >
> > ### 4. **Video Scene Classification (Weaknesses)**
> >
> > We appreciate your concern regarding the relevance of time-aware embeddings for scene classification. While scene classification is generally time-invariant, temporal cues (e.g., lighting changes) can provide useful information for certain outdoor scenes.
> >
> > In addition, thank you for notifying us about the gap in Hollywood2-Scene performance results of VideoMAE + CLIP we reached at the original learning rate. After some further trials with lower learning rates, we obtained new reasonable results. Below are the summarized training configurations and results:
> >
> > | Dataset    | Learning Rate | Epochs | Batch Size | Acc (VideoMAE + Salem et al. 2022) | Acc (VideoMAE + Zhai et al. (2019)) | Acc (VideoMAE + CLIP) | Acc (VideoMAE + TICL) |
> > | ---------- | ------------- | ------ | ---------- | ---------------------------------- | ----------------------------------- | --------------------- | --------------------- |
> > | Hollywood2 | 1e-4 → 5e-5   | 20     | 2          | 32.99% → 45.53%                    | 32.65% → 51.03%                     | 22.51% → 52.92%       | 59.79%→56.53%         |
> >
> > We will update all the affected results accordingly in the manuscript.
> >
> > ### 5. **Time-Based Editing (Weaknesses)**
> >
> > Your suggestion to evaluate shadow realism and color hues in time editing is compelling.
> >
> > - While our current editing pipeline demonstrates the feasibility of temporal attribute manipulation, it is not explicitly optimized for physically accurate outputs. We are not sure whether our method is able to handle this challenging editing task, but we would like to consider this task in future work.
> >
> > We sincerely appreciate your thoughtful review and suggestions, which have been instrumental in refining our work. We are committed to addressing these points comprehensively in the revised manuscript.
> >
> > Thank you once again for your valuable feedback.

---

### Official Review · Reviewer_EuJG · 2024-11-04

**Soundness:** 3
**Presentation:** 4
**Contribution:** 2
**Rating:** 5
**Confidence:** 5

**Summary:**

The paper addresses the problem of predicting the hour from a given image, leveraging a contrastive loss framework similar to CLIP to align visual space (image) with time representation. It also proposes a data cleanup method and introduces the TOC dataset for hour prediction. The effectiveness of its learned representations is evaluated across multiple downstream tasks, including retrieval, scene classification, and time-aware image editing.

**Strengths:**

- The paper offers solid motivation with clear writing and well-designed diagrams, facilitating comprehension. Additionally, the visualizations effectively illustrate the method's potential.

- Baseline comparisons are well-chosen, including standard regression methods and varied model architectures such as CLIP, DINOv2, and ConvNext, providing a extensive evaluation.

- The paper presents a broad range of applications, including generative tasks. The choice of diverse experiments is commendable.

**Weaknesses:**

- **W1**: A key limitation of the proposed method is its limited technical contribution. The approach employs a simple MLP to project hour-based one-hot encodings into the representation space without advancing time representation in a meaningful way. For instance, while GeoCLIP introduces Random Fourier Features (RFF) to encode geolocation effectively, this work lacks a specific contribution in time representation, appearing more as a direct adaptation of GeoCLIP for time embeddings.
- - **W1.1**: Specifically, the method encodes the floating-point hour value into a one-hot representation of discrete classes (Appendix A.3). This approach underutilizes the ground-truth data, reducing precise time information into approximate class categories. An improved approach might represent time in a hierarchical manner—for instance, with a top-level division for the quarter of the day, followed by classifiers for each hour and even down to minute level—thus preserving the granularity of the original data.

- **W2**: Another major concern is the limited scope of the proposed method. Since it addresses only hour information, it does not account for other factors that significantly affect visual similarity. For example, the time of year (season) can substantially alter a location's appearance, making the problem ill-defined without considering month information. Another influencing factor could be the geographic location, which also impacts visual appearance.

- **W3**: The details provided about the TOC dataset in the Appendix (particularly Fig. 9) reveal a clear skew towards countries in the Western and Northern hemispheres. This imbalance is undesirable for the proposed hour-prediction problem, as geolocation significantly impacts appearance-based similarity in relation to time representation.

- **W4**: On closer examination of the time-based editing results (Fig. 28), it’s apparent that the generated edits fails to retain original image information. For instance, in Fig. 28(b), second row, the building structure noticeably changes. It may not be clear at the low-resolution results provided in the paper. Although this is an observation not central to my evaluation, such shifts may defeat the intended purpose of the editing application.

- **W5**: The video scene classification task raises two questions:

- - How does this task contribute to evaluating time-aware representations? Scene classification should ideally be time-invariant.

- - The proposed scene classification pipeline does not look intuitive. For zero-shot classification, it would be intuitive to use only the candidate model (e.g., CLIP) features. The introduction of VideoMAE here is unexpected, and it would be helpful for the authors to clarify this choice in the rebuttal.

**Questions:**

Please refer to the limitations section for further details.

Q1: The choice of partitioned classes for representing time is unclear. Could the authors provide a justification for this design choice over other choices like using RFF or using an hierarchical representation? (See Weakness W1.)

Q2: How is video scene classification a relevant downstream application for evaluating hour-aware representations?

Q3: What is the reason behind appending VideoMAE features? Could the authors provide results, such as those in Table 3, without the inclusion of VideoMAE features?

**Details Of Ethics Concerns:**

**Minor concern**: The details provided about the proposed TOC dataset in the Appendix (particularly Fig. 9) reveal a clear skew towards countries in the Western and Northern hemispheres. This imbalance is undesirable for the proposed hour-prediction problem, as geolocation significantly impacts appearance-based similarity in relation to time representation.

---

> ### Author Response · Authors · 2024-11-14
> **Response part 1**
>
> Dear Reviewer EuJG,
>
> Thank you for your thoughtful feedback and constructive comments on our work. We greatly value your suggestions, and we hope to address your concerns below.
>
> ### 1. **Technical Contribution (W1, W1.1, Q1)**
>
> We appreciate your concerns regarding the simplicity of our MLP-based projection module and the one-hot encoding approach for time representation. Our primary motivation was to prioritize a lightweight design that adapts well to diverse downstream tasks.
>
> - We tested alternative time encodings, including **Random Fourier Features (RFF)** and **T2V**, which we have included following rows expanding **Table 2** with additional ablations for using the **RFF** (input is hour and minute) and **T2V**. The results confirm that our design achieves better performance while maintaining simplicity and adaptability for downstream tasks.
>
> | Image Encoder       | $f_{\theta_t}$ | $f_{\theta_{ITA}}$ | TOC Test Set: Top-1 Acc (%) ↑ | TOC Test Set: Top-3 Acc (%) ↑ | TOC Test Set: Top-5 Acc (%) ↑ | TOC Test Set: Time MAE (min) ↓ | AMOS Test Set: Top-1 Acc (%) ↑ | AMOS Test Set: Top-3 Acc (%) ↑ | AMOS Test Set: Top-5 Acc (%) ↑ | AMOS Test Set: Time MAE (min) ↓ |
> | ------------------- | -------------- | ------------------ | ----------------------------- | ----------------------------- | ----------------------------- | ------------------------------ | ------------------------------ | ------------------------------ | ------------------------------ | ------------------------------- |
> | **CLIP (ViT-L/14)** | RFF            | ✓                  | 16.75                         | 65.14                         | 46.61                         | 206.50                         | 6.07                           | 15.78                          | 22.27                          | 290.70                          |
> |                     | T2V            | ✓                  | 17.70                         | 45.69                         | 66.11                         | 185.89                         | 7.37                           | 21.74                          | 35.10                          | 264.25                          |
> |                     | Ours           | ✓                  | 20.61                         | 49.01                         | 67.83                         | 171.65                         | 13.55                          | 38.50                          | 57.28                          | 187.87                          |
>
> Regarding your suggestion for hierarchical time representations, we appreciate this suggestion with interesting insights, however, we would like to clarify a few points about it:
>
> - Firstly, we have discussed how different class partitioning granularities marginally contributes to the benchmark metrics in appendix A.3 (more specifically: Figure 15, 16, 17). We did not observe promising results motivated us stacking the models with different granularities to improve timestamp estimation.
>
> - Secondly, our focus was on establishing a robust and efficient baseline for hour prediction model having time-of-day awareness instead of the benchmark themselves.  We will consider it in future iterations of this work.
>
> ### 2. **Scope and Broader Context (W2, Q2)**
>
> We acknowledge that hour prediction alone does not account for broader factors like seasonality or geographic location, which also influence visual appearance.
>
> - However, we deliberately focused on hour prediction as a first step to isolate and study temporal cues in images. This is an underexplored dimension that complements existing work on geolocation and scene understanding. We will revise the manuscript to better articulate these design choices and explicitly acknowledge this limitation.
>
> Regarding video scene classification:
>
> - While scenes may appear to be time-invariant, temporal information can offer subtle cues (e.g., lighting, activity patterns) that influence classification because some scenes have are conceptually related to time as we have covered in Sec. 5.3.2 (line 423-460). A.7.2(line 1496-1593) and A.9 (line 1751-1755).
>
> ### 3. **Inclusion of VideoMAE Features (W5, Q3)**
>
> The inclusion of VideoMAE features was aimed to stabilize training in videos. However, we understand that this addition may appear inconsistent with our focus on static temporal embeddings. After a few further experiments on linear probing, we find that the models can still produce meaningful results which we will update in future versions.
>
> Continued...

---

> > ### Author Response · Authors · 2024-11-14
> > **Response part 2**
> >
> > ### 4. **Dataset Imbalance (W3)**
> >
> > We acknowledge the geographic skew in the TOC dataset. This is due to the original imbalance of the Flickr user group and possibly persists in the whole internet image data distribution. We understand your concern on the skew, however, it's not very reasonable to remove the excessive part of western and northern hemisphere images just to make sure it distributes equally across the globe which does not align with reality.
> >
> > ### 5. **Time-Based Editing Results (W4)**
> >
> > We appreciate your observation regarding artifacts in the time-based editing results. These artifacts stem from the simple latent optimization editing baseline we used, which is not optimized for structural preservation. While the shape change is a limitation, the results demonstrate our primary focus: the ability to modify temporal attributes.
> >
> > As shown in Figure 29 (Appendix A.8.2, line 1674-1724), stronger editing baselines paired with our time-aware loss produce significantly better results. We will clarify this distinction in the manuscript.
> >
> > We sincerely thank you for your constructive feedback, which has helped improve our work. We look forward to further strengthening our manuscript based on these insights.

---

### Note · Authors · 2024-11-14

I have read and agree with the venue's withdrawal policy on behalf of myself and my co-authors.